# Evolutionary and functional history of the *Escherichia coli* K1 capsule

Sergio Arredondo-Alonso[1,2], George Blundell-Hunter [3,20], Zuyi Fu[4,20], Rebecca A. Gladstone[1,2,20], Alfred Fillol-Salom[4], Jessica Loraine[3], Elaine Cloutman-Green[5], Pål J. Johnsen [6], Ørjan Samuelsen [6,7], Anna K. Pöntinen [1,7], François Cléon[6], Susana Chavez-Bueno [8,9], Miguel A. De la Cruz [10,11], Miguel A. Ares[10], Manivanh Vongsouvath[12], Agnieszka Chmielarczyk[13], Carolyne Horner[14], Nigel Klein [5], Alan McNally [15], Joice N. Reis[16,17], José R. Penadés [4], Nicholas R. Thomson [2,18], Jukka Corander [1,2,19] ✉, Peter W. Taylor[3] ✉ & Alex J. McCarthy [4] ✉

*Escherichia coli* is a leading cause of invasive bacterial infections in humans. Capsule polysaccharide has an important role in bacterial pathogenesis, and the K1 capsule has been firmly established as one of the most potent capsule types in *E. coli* through its association with severe infections. However, little is known about its distribution, evolution and functions across the *E. coli* phylogeny, which is fundamental to elucidating its role in the expansion of successful lineages. Using systematic surveys of invasive *E. coli* isolates, we show that the K1-*cps* locus is present in a quarter of bloodstream infection isolates and has emerged in at least four different extraintestinal pathogenic *E. coli* (ExPEC) phylogroups independently in the last 500 years. Phenotypic assessment demonstrates that K1 capsule synthesis enhances *E. coli* survival in human serum independent of genetic background, and that therapeutic targeting of the K1 capsule re-sensitizes *E. coli* from distinct genetic backgrounds to human serum. Our study highlights that assessing the evolutionary and functional properties of bacterial virulence factors at population levels is important to better monitor and predict the emergence of virulent clones, and to also inform therapies and preventive medicine to effectively control bacterial infections whilst significantly lowering antibiotic usage.

Bacteria across a wide range of phyla produce capsular polysaccharides that are associated with diverse experimentally validated functions and have been shown to improve bacterial persistence and adaptation to new environments[1,2]. Human bacterial pathogens use these capsules as major virulence determinants to promote colonization and persistence in the gastrointestinal, respiratory, and urogenital tracts, as well as other tissues[3–5]. Capsular polysaccharides are often identical or similar in structure and chemical composition to polysaccharides found in human tissues, thereby providing non-immunogenic coatings for bacteria to survive in human tissues and to cause disease[5]. In addition, capsules can reduce the efficiency of antimicrobial peptides and complement[6–9], suppress phagocytosis by innate immune cells and promote intracellular survival[10–17], and contribute to defence against antimicrobial agents[8,18]. Although *E. coli* can produce around 80 distinct capsular chemotypes that are organised into four major groups[19], only a subset of these chemically distinct capsular types are associated with the capacity to cause invasive extraintestinal diseases; such infections include bloodstream

infections (BSI), pyelonephritis and meningitis[20]. In particular, the polysialic acid-containing K1 capsule[21,22], chrondodontin-containing K4 capsule[23] and heparosan-containing K5 capsule[24,25] are associated with the extraintestinal pathogenic *E. coli* (ExPEC) clones linked to such invasive diseases[26,27], likely through the propensity of these capsule type to mimic polysaccharides present on cells in the human tissues[5]. However, the epidemiology of the capsular types remains largely unexplored due to the absence of serological typing data or specific methods that can computationally predict *E. coli* capsular types based on whole-genome sequencing data. Our current understanding of the evolution and functional properties of the distinct capsular polysaccharides in the global *E. coli* population is therefore limited and mainly based on pre-genomic studies.

The K1 capsule polysaccharide has repeatedly been linked to BSI, neonatal meningitis and pyelonephritis[5,28–31]. The K1 capsule is a homopolymer of α−2,8-linked N-acetylneuraminic acid (sialic acid; NeuNAc) termed polysialic acid (polySia) that mimics the polySia modification found on human neuronal and immune cells[32–35] and likely promotes the capacity of K1-expressing *E. coli* to hide and reside within the blood and neuronal compartments. Indeed, polySia prevents full activation of innate host defences and confers resistance to complement- and phagocyte-mediated killing[36–38]. In agreement with epidemiological links of K1 capsule to invasive human infections, experimental animal models using isogenic strains have revealed that K1 expression promotes stable gastrointestinal (GI) tract colonisation and promotes the development of invasive systemic infections by *E. coli*[39–42]. In contrast to the association of K1 with BSI and meningitis, other well-studied K antigens including types K2, K4 and K5 are mostly associated with UTIs[5,6,43,44]. Despite the association of K1 encapsulated *E. coli* with BSI and meningitis, and the fact that polySia can be utilized as a powerful tool for diagnosis and therapeutic targeting[45,46], we lack basic knowledge on the prevalence, evolution and functional properties of the K1 capsule at the population level. This lack of knowledge limits our capacity to develop efficient strategies to combat *E. coli* infections.

In this study, we elucidated the prevalence, distribution, and evolution of the K1 capsule in *E. coli* populations by considering a global dataset of 5065 clinical isolates. The position and synteny of the K1 capsule (K1-*cps*) locus in the genomes were resolved by considering K1 complete genomes. Using a Bayesian inference approach, we estimated the introduction date of K1-*cps* in the most predominant extraintestinal lineages. To show that the K1 capsule was functional among the distinct lineages, we performed several phenotypic assays to show that the K1 capsule was expressed and conferred *E. coli* with immune resistance independent of its genetic background. Our results show that a quarter of BSIs are caused by *E. coli* carrying this the K1-*cps* locus, and this is driven by multiple introductions of K1-*cps* into the ExPEC pathotype. For the first time, we estimated the introduction times of the capsule among the main ExPEC lineages and dated that the K1-*cps* locus was acquired at least 500 years ago. In support of the role of K1 capsule in virulence, we show that the enzymatic removal of the K1 capsule renders *E. coli* susceptible to complement-replete human serum, suggesting that the therapeutic use of capsule depolymerases is likely to be a promising approach for the prevention and treatment of these infections.

## Results

### A quarter of the ExPEC population has the K1-*cps* locus

Despite the association of K1-*cps* to the ExPEC pathotype since the 1980s[21,47–49], and the availability of whole-genome sequencing data for ExPEC isolates, the epidemiology of the K1 capsular type has remained largely unexplored in the post-genomic era. The K1 capsule biosynthesis locus belongs to group 2 capsules[26], and is composed of eight genes in two conserved regions (regions 1 and 3) shared between all group 2 capsule types and an additional 6 genes (region 2) unique to the K1-*cps* locus (Fig. 1a).

To estimate the prevalence of the K1-*cps* locus among ExPEC isolates, we assessed two unbiased longitudinal studies, NORM[50] and BSAC[51], that characterized BSIs in Norway ($n = 3254$) and United Kingdom ($n = 1509$), respectively. As the BSI isolates in both studies were collected regardless of their clonal background, antimicrobial resistance profile or other bacterial phenotypic or genotypic characteristics, the NORM and BSAC collections offer a representative survey of BSI clones that have circulated in comparable host populations during the timespan of the studies. These datasets therefore provide a valuable platform to estimate the K1-*cps* prevalence in *E. coli* BSI populations. The population prevalence of the K1-*cps* locus among BSI isolates was estimated to be 24.0% and 22.9% for the NORM and BSAC collections, respectively (Table 1). The finding that a similar proportion of BSI isolates are positive for K1-*cps* in the two independent unbiased longitudinal studies provides strong evidence that the K1 is linked with *E. coli* propensity to cause BSIs.

Group 2 capsules, including K1, are classically assumed to be expressed in *E. coli* isolates causing extraintestinal infections, but not in *E. coli* causing diarrhoeal diseases[26]. To clarify whether the K1-*cps* is associated with any other *E. coli* pathotypes, we analysed the Horesh et al. collection that consists of a comprehensive, high-quality and pathotype-defined collection of *E. coli* genomes[52]. We specifically screened for the presence of the K1-*cps* locus in the 5,236 diarrhoeagenic isolates from the Horesh et al. collection that includes the pathotypes (i) enteropathogenic *E. coli* (EPEC), (ii) enterotoxigenic *E. coli* (ETEC), (iii) enterohaemorrhagic *E. coli* (EHEC), (iv) enteroaggregative *E. coli* (EAEC), (v) enteroinvasive *E. coli* (EIEC), (vi) diffusely adherent *E. coli* (DAEC) and (vii) adherent invasive *E. coli* (AIEC). We observed that only 0.1% (5/5236) of the diarrhoeagenic isolates carried the K1-*cps* locus, therefore discarding a role of the K1 capsule in diarrhoeagenic diseases.

Given that ~25% of ExPEC isolates possess the K1-*cps* locus, it is important to analyze the clonality of these isolates to understand the population structure and to define how ExPEC clones with enhanced virulence properties emerge and evolve. We, therefore, examined the distribution of the K1-*cps* locus across the distinct ExPEC phylogroups (phylogroups A, B1, B2, C, D, E and F). For both the NORM and BSAC datasets, the K1-*cps* was frequent in phylogroup B2 (31.7% and 29.4%, respectively) and phylogroup F (50.4% and 23.2%, respectively), but also observed in phylogroups A (9.3% and 6.1%, respectively) and D (0.6% and 1.4%, respectively). We did not detect K1-*cps* in phylogroup B1, C or E. The distribution of the K1-*cps* locus among clonal complexes (CC, defined here as belonging to a lineage named after the dominant sequence type) was almost identical for both NORM and BSAC collections (Table 1). Within phylogroup B2, the K1-*cps* was mostly present in CC95 followed by CC141, CC144, and CC80 (Table 1). In addition, we observed the capsule in a few isolates of the pan-susceptible clade B of CC131, but not in the multidrug-resistant (MDR) C1 and C2 clades of CC131. The K1-*cps* locus was also present in the CC59 and CC62 within phylogroup F and in CC10 within phylogroup A (Table 1). This data indicates that 22–24% of all BSI isolates are not caused by a single successful K1-*cps*+ ExPEC clone, but by multiple distinct K1-*cps*+ clones.

To analyze the distribution and evolution of the K1-*cps* across a global collection of ExPEC isolates, we expanded the NORM and BSAC studies by incorporating isolates from other countries, time periods and sources that were genotypically or phenotypically detected as K1-*cps*+ or K1+, respectively. This batch of samples included: (i) newly generated adult and neonatal samples phenotypically detected as K1+, $n = 201$, from six countries (Brazil, Laos, Mexico, Poland, United Kingdom, United States), (ii) samples carrying the K1-*cps* locus from the pre-antibiotic era (from 1932 onwards) that form part of the Murray collection, $n = 15$[53] and (iii) samples carrying the K1-*cps* locus from the

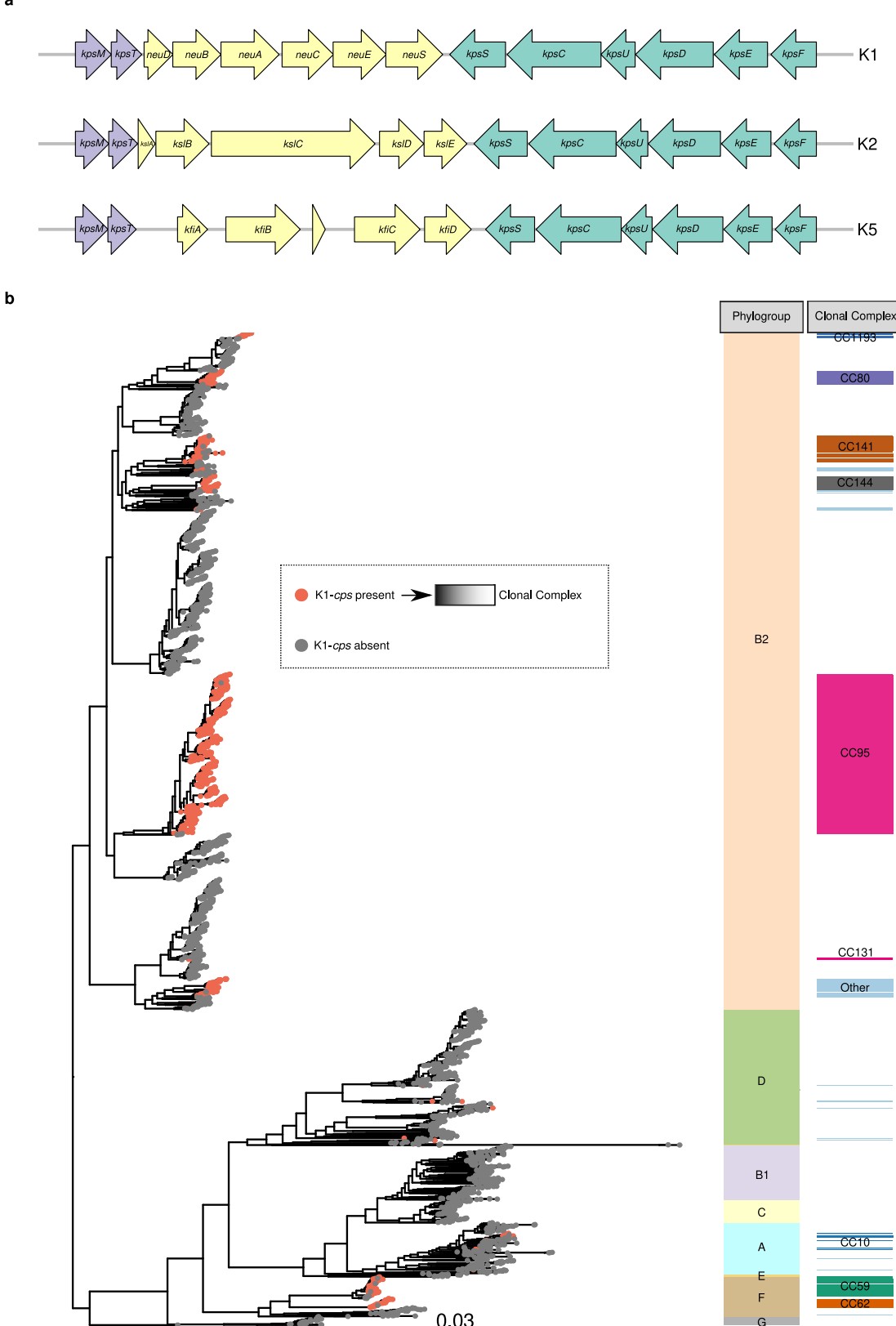

Horesh collection, $n = 86$[52]. We collated all genome isolates (5065) and generated a maximum-likelihood phylogeny. A total of 1427 isolates possessed the K1-*cps* locus (Fig. 1b). The phylogeny overlaid with the metadata (also available at Dataset S1) can be interactively queried in the following Microreact project https://microreact.org/project/cm8f8adPyWoQqemAAgvrmi-k1-context. For isolates in which the complete K1-*cps* locus, from *kpsM* to *kpsF* (Fig. 1a), was assembled in one contig (1409/1427, 98.7%), the product of *kpsT* showed the most conserved protein sequence with 98.7% (1391/1409) of isolates carrying an identical amino acid sequence (Supplementary Fig. 1). In contrast, *neuE*, the most diverse gene in the locus, displayed a total of 76 distinct variants (Supplementary Fig. 1).

**Fig. 1 | K1-*cps* genomic organisation and distribution in the *E. coli* global dataset. a** Genetic organisation of the group 2 capsule locus in *E. coli* encoding K1, K2 and K5 antigens. Regions 1 and 3 (in green and purple, respectively) are responsible for capsule export and are shared among K antigens belonging to group 2 capsules. Region 2 (in yellow) is serotype-specific and involved in capsule biosynthesis. The coding sequences (CDs) are represented as arrows in the diagram and are labelled according to their gene names. **b** Maximum-likelihood phylogenetic tree of 5065 *E. coli* clinical isolates. The phylogeny was reconstructed by mapping all isolates against the *Escherichia coli* strain EC958 (accession HG941718; https://www.ncbi.nlm.nih.gov/nuccore/HG941718.1), using the fast and GTR model

provided in iqtree. The scale bar corresponds to the expected number of nucleotide substitutions per site. In the phylogeny, nodes were coloured, based on the absence (grey tip) or presence (red tip) of the K1-*cps* locus. Next to the phylogeny, the metadata block, Phylogroup, indicates the phylogroup of all isolates regardless of possessing or not the K1 capsule of the isolates bearing the K1-*cps* gene cluster. The metadata block, Clonal Complex, indicates the clonal complex of the isolates possessing the K1-*cps* locus in their genome. The phylogeny and metadata can be interactively queried using the following Microreact link https://microreact.org/project/cm8f8adPyWoQqemAAgvrmi-k1-context. Source data are provided as a Source Data file.

## The dominant ExPEC lineage CC95 acquired the K1-*cps* locus over 250 years ago

CC95 is one of the most dominant ExPEC clones associated with community-onset and nosocomial infections worldwide[20], and the K1-*cps* locus is ubiquitous in invasive isolates of this lineage (99.8–100% of CC95 isolates are K1-*cps*+; Table 1). We hypothesized that the successful expansion and establishment of the K1-*cps*+ CC95 clone in the *E. coli* population has been driven by a single acquisition of the K1-*cps* locus and clonal expansion, rather than multiple independent K1-*cps* acquisitions by CC95. We further investigated the genetic context and evolution of the K1-*cps* in the lineage by obtaining a dated phylogeny of CC95 NORM genomes to determine the most common recent ancestor (tMRCA) for K1-*cps* within CC95, and by unambiguously characterizing the position and gene synteny of the K1-*cps* locus in the *E. coli* genome by retrieving all CC95 genomes with an associated RefSeq complete genome (*n* = 44). Nearly all NORM genomes (441/442) from CC95 (Table 1) possessed the K1-*cps* suggesting that this locus was introduced by its most common recent ancestor (MRCA).

To determine the origin of CC95 (tMRCA), we used BactDating that makes use of a Markov Chain Monte Carlo (MCMC) model to perform Bayesian inference and produce a dating phylogeny. We estimated the origin of CC95 (tMRCA) and thus estimated the introduction of the K1-*cps* locus to be approximately around the year 1768 [95% CI, 1721–1806] (Fig. 2a). In Dataset S2, we confirmed the MCMC convergence of the model as shown by (i) the effective sample sizes of the parameters and (ii) the Gelman and Rubin's convergence diagnostic. In addition, we show a significant temporal signal by comparing the resulting model against a model with equal sampled dates using the deviance information criteria (DIC) (Dataset S2). Thus, the high-frequency of K1-*cps* in CC95 is readily explained by a single acquisition event that occurred approximately over 250 years ago that subsequently spread worldwide as a single clone.

The K1-*cps* locus was present next to the tRNA-*pheV* site, which is localized around the position ~800,000 of the chromosome (matching

the position described in pre-genome literature as 67 min in the *E. coli* genome[54,55]) as revealed by analysis of the RefSeq CC95 complete genomes (Dataset S3). Upstream of the K1-*cps* locus, gene synteny is highly conserved and is characterized by the presence of a type II secretion system[56]. Downstream of the locus, two predominant genome configurations were defined across CC95 (i) the insertion of a pathogenicity island (PAI) that has resulted in the truncation of the tRNA *pheV* gene (26/44, 59%), and (ii) the absence of the PAI and an intact *pheV* gene (16/44, 36%), while 2 genomes (2/44, 5%) showed each one a distinct gene synteny as determined by Panaroo. In the NORM CC95 isolates, we could detect the PAI in 80.27% (354/441) of the CC95 isolates based on the truncation of the *pheV* and presence of the *intB* gene, while 15.87% (70/441) of isolates did not carry the PAI, and in 3.85% of the cases (17/441) we could not determine the presence/absence of the PAI due to fragmented short-read assemblies (Fig. 2a).

This PAI had an average size of 52.4 kbp and was previously termed PAI-V (Fig. 2b)[57,58]. Because PAIs carry one or more accessory genes that encode virulence factors that often function as adhesins, iron-acquisition systems, host defence mechanisms or toxins[59], the acquisition and loss of this PAIs is likely to impact phenotypes and infectivity. We did not observe any trend in the association between the presence/absence of the PAI and the isolation source (BSI vs UPEC) of these 44 complete genomes (Dataset S3), indicating that acquisition or loss of this PAI is not exclusively linked to either type of invasive infection. The mosaic structure and flexible pool of virulence genes carried by PAIs could mean that a certain virulence gene is associated with the PAI in the CC95 lineage. However, a comparison of the virulence genes of the isolates that lost the PAI with the closest PAI+ isolate in the phylogeny indicated the main difference resided in the presence and absence of *papGII* locus encoding the P fimbriae and an iron-acquisition (*ireA*) locus. Interestingly, these genes have been recently identified as key features distinguishing invasive from non-invasive UPEC[60]. Additional genes that have roles in promoting bacterial survival or virulence were not identified on the PAI in CC95 isolates.

Based on the CC95 dated phylogeny (Fig. 2a), we determined that the MRCA did not possess the PAI and that it is likely that there were two introductions of a PAI downstream of the *kpsF* gene in this lineage. The PAI was estimated to have been introduced between 1823 [95% CI, 1789–1851] and 1840 [95% CI, 1809–1864], and then subsequently has become fixed in the genome and predominated through clonal expansion. Interestingly, we estimated a minimum of 14 excision events of the PAI linked to recombination between the two direct short repeats at *pheV* (22 bp)[61] that resemble *att sites* in a process likely mediated by the *intB* P4-like integrase gene[58,62]. This suggests that the PAI can be excised and lost from the *E. coli* genome. In support, this PAI can form a circular excision product that may be exported by larger mobile genetic elements (MGEs)[61]. We also observed a minor and independent introduction of an island downstream of the *kpsF* gene estimated to happen between 1864 [95% CI, 1832–1892] and 1904 [95% CI, 1878–1927] (Fig. 2b). Collectively, this data indicates that the introduction of the K1-*cps* locus in CC95 precedes the acquisition of the PAI, but that accessory genes encoded on the PAI could additionally contribute to the success of CC95 as an ExPEC clone.

## Table 1 | Prevalence of the K1-*cps* locus in the NORM and BSAC collections

| Clonal Complex (CC) | Phylogroup | K1-*cps* prevalence per CC in the NORM collection | K1-*cps* prevalence per CC in the BSAC collection |
|---|---|---|---|
| All | – | 780/3254 (24.0%) | 345/1509 (22.9%) |
| CC95 | B2 | 441/442 (99.8%) | 189/189 (100%) |
| CC141 | B2 | 78/89 (87.6%) | 21/24 (87.5%) |
| CC59 | F | 48/58 (82.8%) | 16/22 (72.7%) |
| CC144 | B2 | 38/38 (100%) | 23/23 (100%) |
| CC80 | B2 | 41/41 (100%) | 22/22 (100%) |
| CC1193 | B2 | 21/25 (84.0%) | 0/1 (0%) |
| CC62 | F | 19/19 (100%) | 19/20 (95.0%) |
| CC10 | A | 12/67 (17.9%) | 7/40 (17.5%) |
| CC131 | B2 | 4/203 (2.0%) | 1/202 (0.5%) |
| Other | – | 78/2272 (3.4%) | 47/966 (4.9%) |

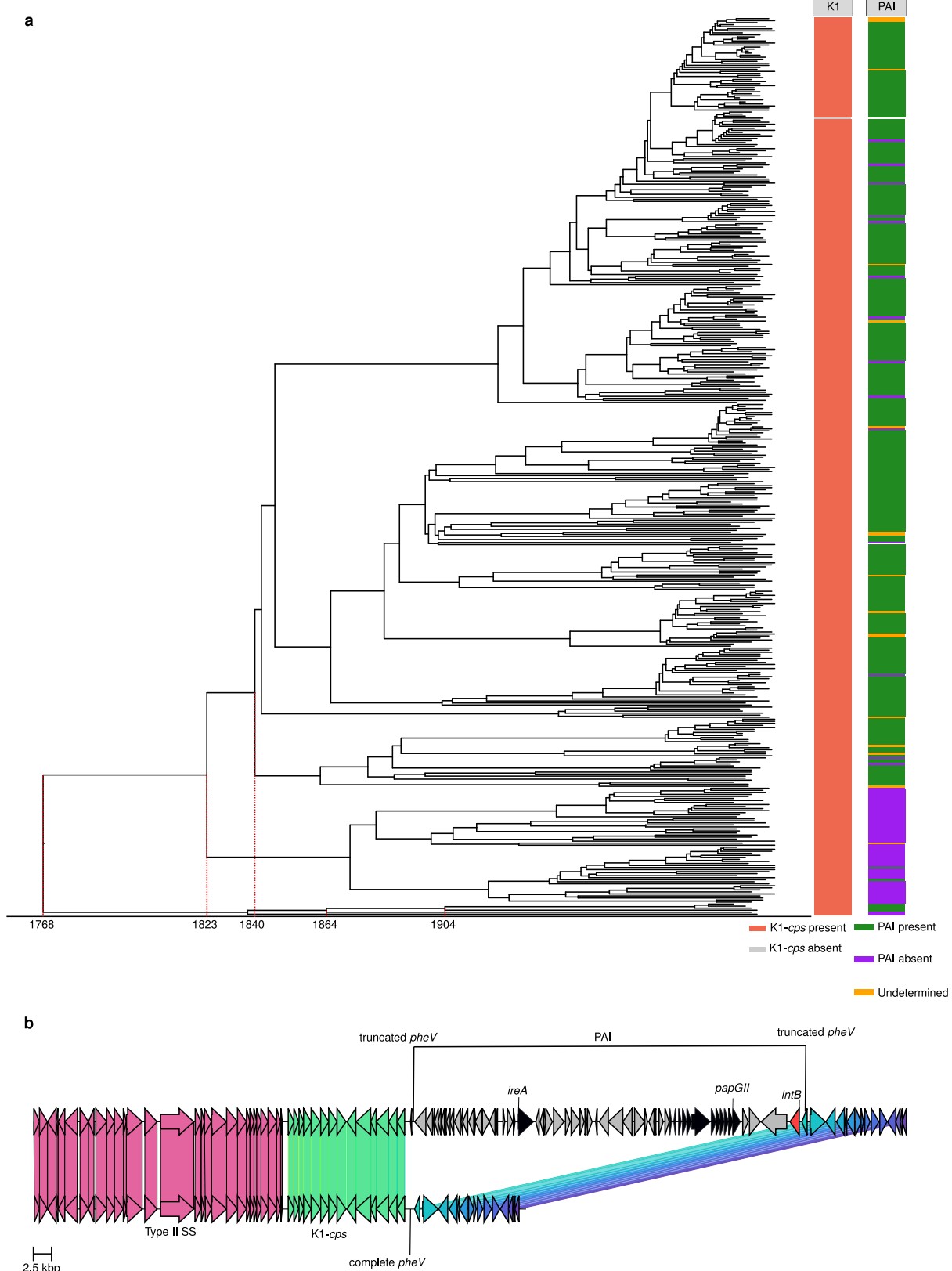

### Multiple K1-*cps* locus acquisitions across the major ExPEC lineages

Given that the K1-*cps* + CC95 clone emerged over 250 years ago and is globally successful, we hypothesized that the CC95 population is likely to have acted as the donor of the K1-*cps* locus during the emergence of other K1-*cps* + ExPEC clones in other phylogroups. For each K1-*cps*+

lineage, we performed a hybrid assembly to obtain a complete chromosomal sequence (Dataset S4) that allowed us to infer the gene synteny present downstream of the K1-*cps* locus and shown in Fig. 4.

We first analysed the acquisition of the K1-*cps* locus in other phylogroups (CC59 and CC62 in phylogroup F; CC10 in phylogroup A). Surprisingly, the K1-*cps* locus was introduced into CC59 by its MRCA

**Fig. 2 | Downstream gene synteny of the K1-*cps* locus in CC95 complete genomes. a** Dated phylogeny of the CC95 lineage considering as reference genome the RefSeq assembly accession GCF_008632595.1. The K1-*cps* presence (red) and absence (grey) is mapped as a metadata block. Carriage of the pathogenicity island (PAI) is mapped as a metadata block; present (green), absent (purple) or unknown (orange). The ticks on the *x*-axis of the phylogeny indicate: (i) the time of the most recent common ancestor (tMRCA) for the lineage (1768) [95% CI, 1721–1806], (ii) the time interval of the primary PAI acquisition, between 1823 [95% CI, 1789–1851] and 1840 [95% CI, 1809–1864], and (iii) the time interval of a second PAI acquisition, dated between 1864 [95% CI, 1832–1892] and 1904 [95% CI, 1878–1927]. **b** Genetic organisation of the two most predominant synteny blocks observed downstream of the *kpsF* gene for the complete CC95 chromosomal sequences illustrated with the RefSeq assemblies GCA_002285755.1 (top block) and GCA_001280405.1 (bottom block). The top block illustrates the presence of the PAI downstream of the *kpsF* gene which resulted in the truncation of the tRNA *pheV* gene. The P4-like integrase *intB* present in the PAI is coloured red. The virulence genes present in the PAI, *ireA* and the *papGII* operon, are coloured black. The bottom block illustrates the second most predominant synteny block represented by an intact tRNA *pheV* gene and the absence of a PAI downstream the K1-*cps* locus. Source data are provided as a Source Data file.

around the year 1525 [95% CI, 1044–1730] (Fig. 3, Dataset S2). The BactDating model inferring the tMRCA of CC59 showed MCMC convergence of the parameters and a significant temporal signal (Dataset S2). The median number of SNP differences and the SNP profile of the *neu* region (Supplementary Fig. 2) together with the conserved gene synteny observed downstream of the *kpsF* gene (Fig. 4) indicated that CC59 does not share the same K1-*cps* locus with CC95. Instead, CC59 shared the same K1-*cps* locus with CC62, and shared a common ancestor with CC62 within the *E. coli* wide-species tree (Fig. 1). This suggests that the MRCA of these lineages in phylogroup F already carried the K1-*cps* locus. Correspondingly, the data conclusively refute that CC95 was the original donor of the K1-*cps* locus for CC59/CC62, and vice versa, that CC59/CC62 was the original donor(s) of the K1-*cps* locus for CC95.

In CC10 of phylogroup A, K1-*cps* was acquired in two independent events, estimated to have happened between the years: (i) from 1789 [95% CI, 1624–1865] to 1821 [95% CI, 1677–1890] and (ii) from 1853 [95% CI, 1738–1907] to 1873 [95% CI, 1772–1922], respectively (Fig. 3). The BactDating model predicting the date of these events showed MCMC convergence of the parameters and a significant temporal signal (Dataset S2). For one of these events, CC59/CC62 acted as a K1-*cps* donor based on the number and profile of SNP differences (Supplementary Fig. 2), and conserved gene synteny (Fig. 4). Thus, this analysis revealed that CC95 did not act as the donor of K1-*cps* for phylogroup F (CC59, CC62) or phylogroup A (CC10), but instead confirmed the transfer of the K1-*cps* locus occurred from phylogroup F to phylogroup A.

Next, we analysed the acquisition of K1-*cps* by the non-CC95 clones within phylogroup B2. We observed that *E. coli* belonging to CC59/CC62 from phylogroup F acted as K1-*cps* donors for CC1193 from phylogroup B2. We also observed the exchange of the K1-*cps* among lineages from phylogroup B2: (i) between CC95 and CC144, and (ii) between CC141 and CC80. However, we could not estimate the tMRCA of these additional lineages and this prevented us from concluding the directionality of exchange. The analysis further demonstrated that a sub-lineage of the recently globally disseminated lineage CC131 acquired the K1-*cps* locus in a single acquisition estimated to have happened between the years from 1957 [95% CI, 1946–1967] to 1999 [95% CI, 1995–2002] (Fig. 3). In Dataset S2, we show that the CC131 dating phylogeny obtained from BactDating to infer the time of the acquisition showed MCMC convergence of the parameters and a significant temporal signal (Dataset S2). The number of SNP differences with respect to other lineages suggested this lineage has acquired K1-*cps* from a distinct source outside of our collection of sequences (Supplementary Fig. 2). In summary, the emergence of K1-encapsulated ExPEC clones has not been driven by acquisition of the K1-*cps* locus from the globally successful CC95 lineage, despite the high prevalence and close association of this clone with the K1 capsule.

Of note, isolates from lineages in phylogroup A (CC10), phylogroup B2 (CC80, CC131, CC141, CC144, CC1193) and phylogroup F (CC59 and CC62) also frequently had a PAI inserted downstream of the K1-*cps* locus (Fig. 4). Despite high frequency in the PAIs from CC95, the gene encoding IreA was only additionally present in PAI

from CC131, and the gene encoding PapGII was only additionally present in the PAI from lineage CC141. Analysis of gene content of the PAI from isolates of other lineages demonstrates a heterogenous pool of virulence genes. This includes genes encoding characterized virulence factors α-hemolysin toxin (HlyA)[63], cytotoxic necrotizing factor (CNF1)[64], secreted autotransporter toxin (Sat)[65], S-fimbrial adhesin (Sfa)[66], the IrgA homolog adhesin (Ila) homolog adhesin[67] and aerobactin[68]. More comprehensive analysis is required to elucidate the function of PAI-encoded virulence genes as molecular determinants of BSIs.

## K1 capsule expression across diverse ExPEC lineages confers resistance to serum-mediated killing

The K1 capsule is strongly expressed by *E. coli* in blood and is thought to be essential for infection by enhancing resistance to complement-mediated killing[36,69,70], but functional studies have traditionally been performed exclusively in the CC95 genetic background. To test the hypothesis that K1 capsule expression alters the virulence phenotype of *E. coli* independently of the genetic background, we compared the susceptibility of isogenic K1 capsule positive and negative *E. coli* strains to complement C3b deposition and serum killing. We found that inactivation of the *neuC* gene in the K1-*cps* locus enhanced the susceptibility of *E. coli* from three different genetically distinct backgrounds (CC95, CC349 and CC62 from phylogroups B2, D and F, respectively) to complement C3b deposition after incubation in human active serum (Fig. 5a; Supplementary Fig. 3a). Further, inactivation of *neuC* promoted the killing of *E. coli* during incubation in human serum (Fig. 5b; Supplementary Fig. 3a). This data demonstrates that K1 capsule synthesis, independent of genetic background, is an important determinant of *E. coli* survival in human serum, and consequently contributes to virulence.

To functionally validate that the K1 capsule is properly expressed across distinct genetic lineages, and that this expressions results in an altered virulence phenotype of *E. coli*, we evaluated polySia K1 capsule production and the immune evasion properties of 149 *E. coli* isolates that possessed the K1-*cps* locus. We selected isolates that were representative of four phylogroups (*n* = 20 phylogroup A isolates, *n* = 120 phylogroup B2 isolates, *n* = 3 phylogroup D isolates and *n* = 6 phylogroup F isolates). To measure K1 expression we tested the binding of recombinant (r)GFP-labelled EndoNA2 (rEndoNA2-GFP) (Supplementary Fig. 3b), a catalytically inactive form of endosialidase from bacteriophage PK1A2 that specifically targets the polySia K1 capsule containing α2,8-linked N-acetylneuraminic acid units[40,71], to the *E. coli* surface by flow cytometric analysis. rEndoNA2-GFP bound specifically and in a concentration-dependent manner to an K1 capsulated strain (Fig. 5c, Supplementary Fig. 3c). 147/149 (98.6%) isolates from all phylogroups displayed detectable levels of K1 capsule expression (Fig. 5d). This indicates that clinical *E. coli* isolates possessing the K1-*cps* locus properly synthesize polySia at their surface. The heterogeneity in the K1 expression levels could have arisen from variation in the expression of genes for endogenous NeuNAc synthesis in the K1-*cps* locus, a NeuNAc uptake transporter called NanT (encoded by *nanT*) and/or a NeuNAc degradation enzyme called NanA (encoded by *nanA*)[26].

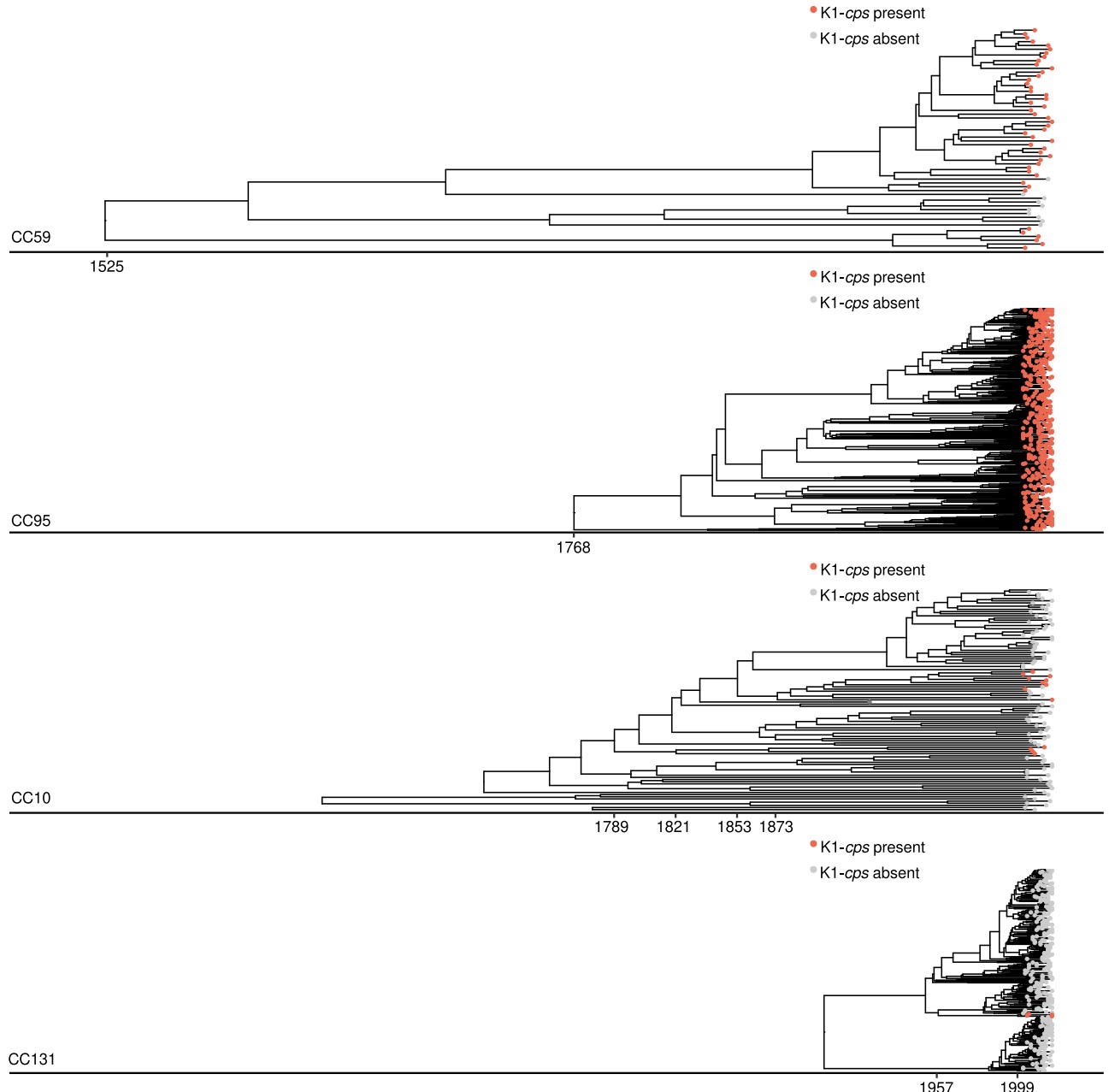

**Fig. 3 | Dated phylogenies and time acquisition of the K1-*cps* locus in the main distinct *E. coli* lineages.** In this analysis, we only included lineages for which we could confidently estimate their tMRCA using BactDating. For each lineage, a tick on the x-axis indicates either the (i) tMRCA (for lineages CC59 and CC95), or (ii) time interval of the K1-*cps* acquisition. For CC59 and CC95, the capsule was introduced by its MRCA and thus their K1-*cps* introductions overlap with the tMRCA (1525 [95% CI, 1044–1730] and 1768 [95% CI, 1721–1806], respectively). For CC10, the

most plausible scenario corresponded to two independent K1-*cps* acquisitions estimated between (i) 1789 [95% CI, 1624–1865] to 1821 [95% CI, 1677–1890] and (ii) 1853 [95% CI, 1738–1907] to 1873 [95% CI, 1772–1922]. For CC131, there was a single introduction of K1-*cps* estimated to have occurred between 1957 [95% CI, 1946–1967] to 1999 [95% CI, 1995–2002]. Source data are provided as a Source Data file.

We next assessed the capacity of the clinical *E. coli* isolates to evade deposition of C3b onto the bacterial surface during incubation in active human serum, using flow cytometric analysis. We found that 144/149 (96.6%) of *E. coli* isolates were resistant to C3b deposition (Fig. 5e). Notably, the level of K1 capsule expression was not correlated with the level of C3b deposition (Supplementary Fig. 3d). Next, we measured the capacity of the *E. coli* isolates to survive in human serum, by quantifying the number of viable bacteria before and after a 3-hour incubation period. Our analysis identified that 138/149 (92.6%) *E. coli* isolates were resistant to serum-killing (Fig. 5f). Like C3b deposition, there was no correlation between K1 capsule expression and percentage of surviving bacteria (Supplementary Fig. 3e).

## Therapeutic targeting of the K1 capsule re-sensitizes diverse ExPEC lineages to the human immune response

The development of effective strategies to prevent and treat *E. coli* diseases is urgent given the rise in hypervirulent and MDR *E. coli* isolates. However, broadly effective vaccines protective against pathogenic *E. coli* and therapeutics that target pathogenic *E. coli* have not been developed[72]. Given that the K1 capsule protects genetically diverse *E. coli* from serum-mediated killing, the removal or reduction of surface K1 capsule is likely to have a major impact on the capacity of *E. coli* to resist the human immune response and to cause infection. Bacteriophage enzymes have been exploited to develop novel strategies to kill bacterial pathogens[73]. This has included utilizing the

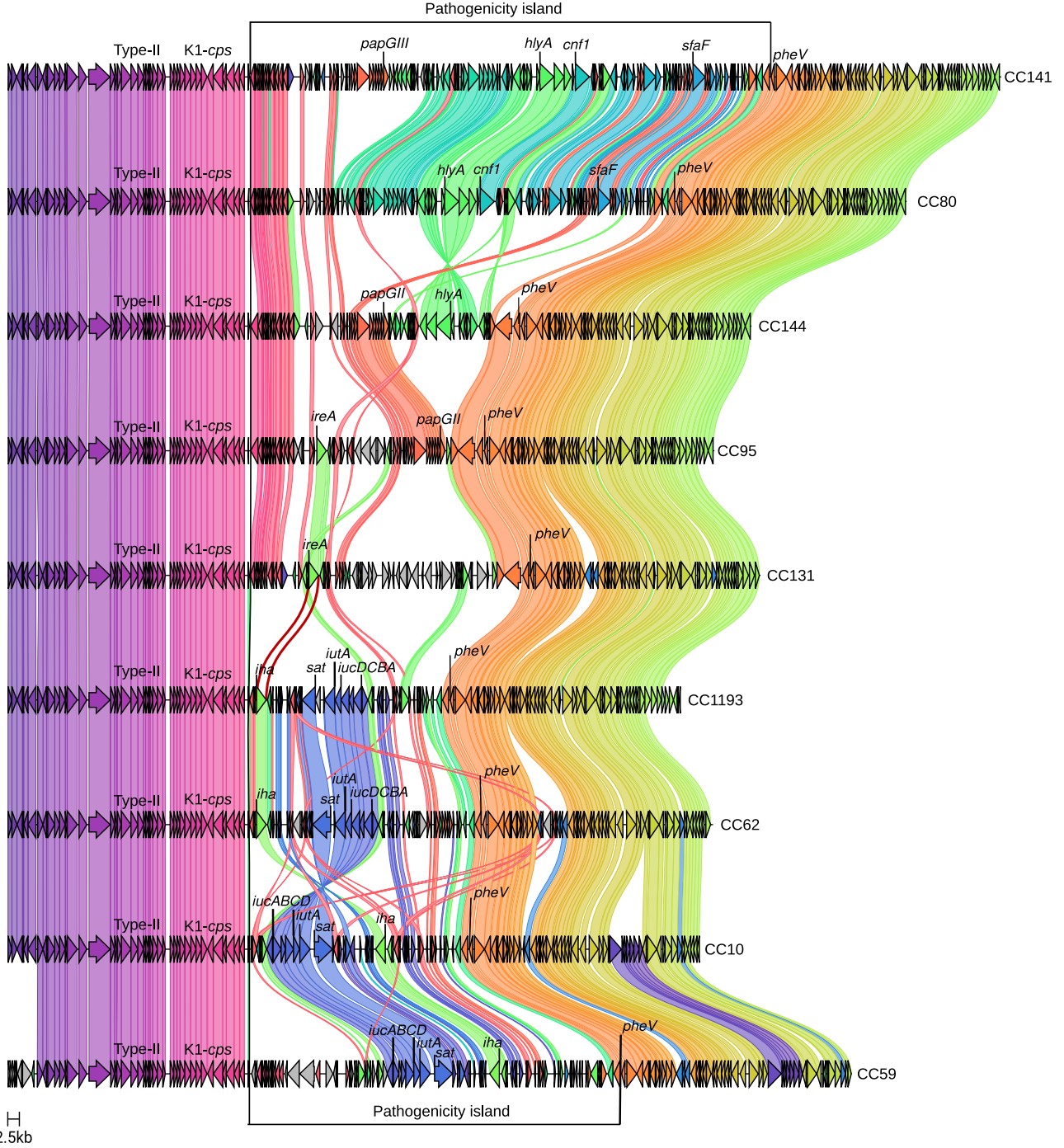

**Fig. 4 | Gene synteny of the neighbouring genomic region of the K1-*cps* locus among the most predominant CCs.** For each lineage, we obtained a chromosomal sequence of the isolates from the NORM collection 30348_1#322 (CC141), 30224_1#187 (CC80), 30224_1#150 (CC144), 30134_6#117 (CC95), 30345_1#279 (CC131), 30348_2#141 (CC1193), 30224_1#376 (CC62), 30348_2#375 (CC10) and 30348_2#365 (CC59). For each chromosomal sequence, we extracted 50 kbp upstream and downstream of the *kpsM* and *intB* genes. *IntB* corresponds to the P4-like integrase gene present and adjacent to the 22 direct-repeat used by the pathogenicity island to be self-mobilised. The position of the tRNA *pheV*, K1-*cps* locus, and type II secretion system is indicated in the plot. The position of the pathogenicity island delimited by the K1-*cps* locus and the truncated tRNA *pheV* is also given. The virulence genes found in the PAI of each lineage are also indicated.

bacteriophage-derived capsule depolymerase EndoE to prevent and treat infections caused by K1 encapsulated *E. coli* in experimental animal models[45,46]. We explored the functional properties of rEndoE and its potential to re-sensitize genetically diverse K1-expressing *E. coli* to human serum.

To assess whether rEndoE was broadly able to re-sensitize genetically distinct K1-expressing *E. coli* to the human innate immune response, we expressed and purified rEndoE (Supplementary Fig. 4a) and examined its capacity to enhance C3b deposition and serum killing of 21 genetically distinct *E. coli* K1-expressing isolates representative of each major phylogroups (A, B2, D and F). Incubation of human serum with 10 µg/mL of rEndoE enhanced the deposition of C3b onto the surface of each K1-expressing *E. coli* isolate during a 30-min incubation period (Fig. 6a; Supplementary Fig. 4b). Pooled analysis of all isolates revealed that rEndoE lead to a 10-fold increase in C3b deposition (Fig. 6b). Moreover, rEndoE decreased the survival of K1-expressing *E. coli* in human serum during a 180-min incubation period (Fig. 6c, d). Notably, as the enzymatic removal of K1 capsule increased

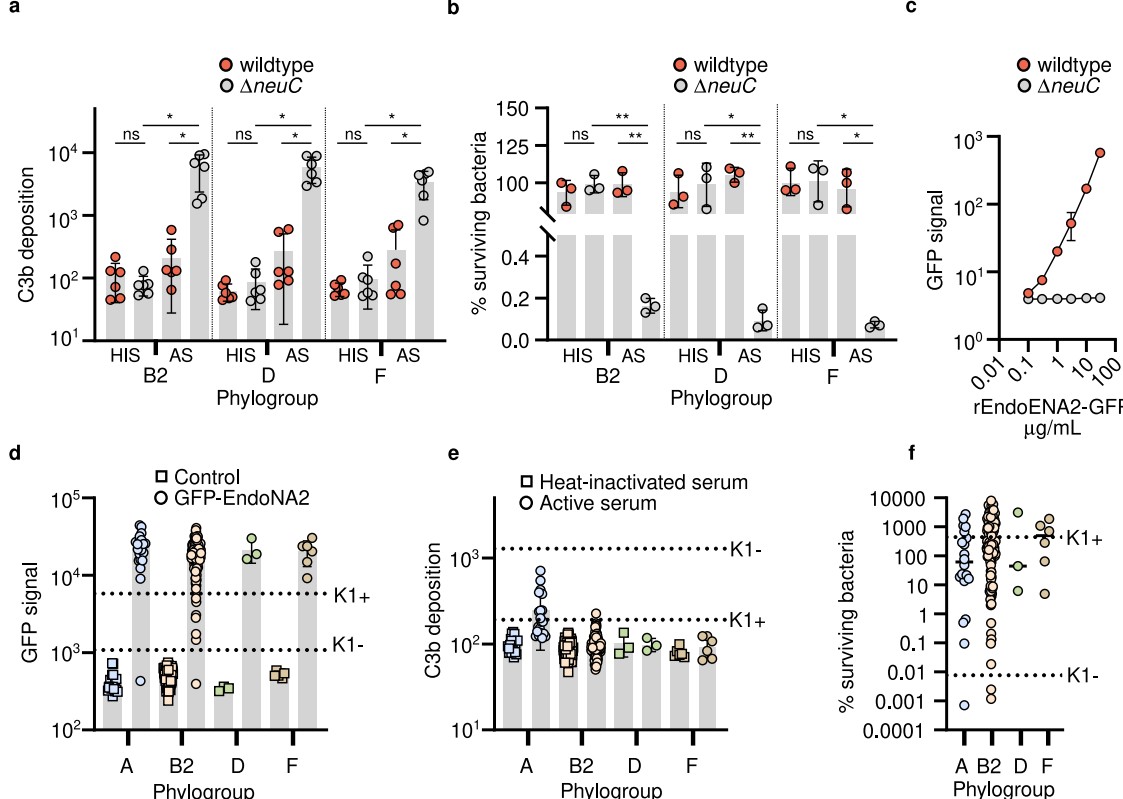

**Fig. 5 | Expression of K1 capsule correlates with resistance of *E. coli* to killing by human serum. a** K1 capsule expression enables *E. coli* strains to resist complement C3b deposition by human serum. Strains were incubated with human serum (AS) or heat-inactivated serum (HIS) for 30 min at 37 °C. Each data point is the mean ± s.d. of $n = 3$ independent experiments. Statistical significance tested by one-way ANOVA with Tukey's post-hoc test (B2, Δ*neuC* HIS vs Δ*neuC* AS *$p = 0.0337$, wildtype AS vs Δ*neuC* AS *$p = 0.0383$; D, Δ*neuC* HIS vs Δ*neuC* AS *$p = 0.0105$, wildtype AS vs Δ*neuC* AS *$p = 0.0105$, F; Δ*neuC* HIS vs Δ*neuC* AS *$p = 0.0146$, wildtype AS vs Δ*neuC* AS *$p = 0.0252$) **b)** K1 capsule expression promotes *E. coli* to resist serum-mediated killing. *E. coli* strains were incubated with human serum or heat-inactivated serum for 3 h at 37 °C. Each data point is the mean ± s.d. of $n = 3$ independent experiments. Statistical significance tested by one-way ANOVA with Tukey's post-hoc test (B2, Δ*neuC* HIS vs Δ*neuC* AS **$p = 0.0019$, wildtype AS vs Δ*neuC* AS **$p = 0.0055$; D, Δ*neuC* HIS vs Δ*neuC* AS *$p = 0.0177$, wildtype AS vs Δ*neuC* AS **$p = 0.0005$, F; Δ*neuC* HIS vs Δ*neuC* AS *$p = 0.0154$, wildtype AS vs Δ*neuC* AS *$p = 0.0141$). **c** Concentration-dependent binding of rEndoENA2-GFP to isogenic K1-*cps*+ and K1-*cps*- (Δ*neuC*) *E. coli* strains, measured by flow cytometry analysis. Each data point is the mean ± s.d. of $n = 3$ independent experiments. **d** Surface expression of polySia K1 capsule by K1-*cps* + clinical *E. coli* isolates, determined by binding of rEndoNA2-GFP to the bacterial surface and subsequent flow cytometry analysis, where mean ± s.d. are displayed. **e** C3b deposition on the surface of K1-*cps*+ clinical *E. coli* isolates after incubation in active human serum, determined by binding of goat ant-human-C3 pAb to the bacterial surface and subsequent flow cytometry analysis, where mean ± s.d. are displayed. **f** Survival of K1-*cps*+ clinical *E. coli* isolates after incubation in human serum, determined by quantification of viable bacteria, where mean ± s.d. are displayed. In (**c**), (**d**), (**e**) and (**f**), *E. coli* isolates are separated by phylogroup, where each data point represents one isolate (the mean quantified from $n = 3$ independent experiments). Total number of isolates in each phylogroup are $n = 20$ for A, $n = 120$ for B2, $n = 3$ for D and $n = 6$ for E. Mean values for isogenic K1+ (wildtype) and K1- (Δ*neuC*) strains are displayed. Source data are provided as a Source Data file.

C3b deposition and serum killing, this demonstrates that K1 capsule provides protection of *E. coli* against the human innate immune response. This observation is noteworthy as the deletion of a gene encoding a major surface component (such as *neuC*) could force a bacterial cell to reconfigure its surface topography, and this could indirectly enhance complement deposition and serum sensitivity. Importantly, our data demonstrates that therapeutic targeting of the K1 capsule can re-sensitize *E. coli* from distinct genetic backgrounds to the human innate immune response.

## Discussion

*E. coli* is one of the most clinically important bacterial pathogens worldwide[74]. Its ubiquitous presence in the human intestinal tract[75], capacity to cause diverse intestinal and extraintestinal infections and the emergence of MDR have made the control of *E. coli* infections a major public health priority[76]. Although there has been a recent focus on the emergence of AMR in *E. coli*[75], an increased incidence of invasive *E. coli* infections has been reported in many settings[77–80] and this trend is likely to continue due to the expanding elderly population. Thus, the identification of the defining features of enhanced virulence in *E. coli*

and the development of novel therapeutics are desirable to inform future control strategies.

The capsule polysaccharide is one of the major virulence determinants in extraintestinal *E. coli* by providing protection against the human immune system[36–38]. Despite the high diversity of K-antigens in *E. coli*[19], we show that the genes encoding K1 capsule synthesis are found in ~25% of all bloodstream isolates from two independent longitudinal studies[50,51]. This high prevalence of the K1-*cps* locus in ExPEC isolates suggests that K1 encapsulated *E. coli* are more invasive compared to other *E. coli* encapsulated with other group 2 capsules. Our combined population genomic analysis coupled with in vitro experiments revealed that the capsule was functionally present in four distinct *E. coli* phylogroups providing immune protection irrespective of the genomic background.

We estimated that the K1-*cps* locus was introduced in the *E. coli* population more than 500 years ago, and we could identify later exchange events of the K1-*cps* locus within and between *E. coli* lineages. This included CC95, a non-MDR lineage responsible for most cystitis, pyelonephritis, bacteraemia, and meningitis cases worldwide[20]. The MRCA of CC95 acquired the K1 capsule around the mid-19th century

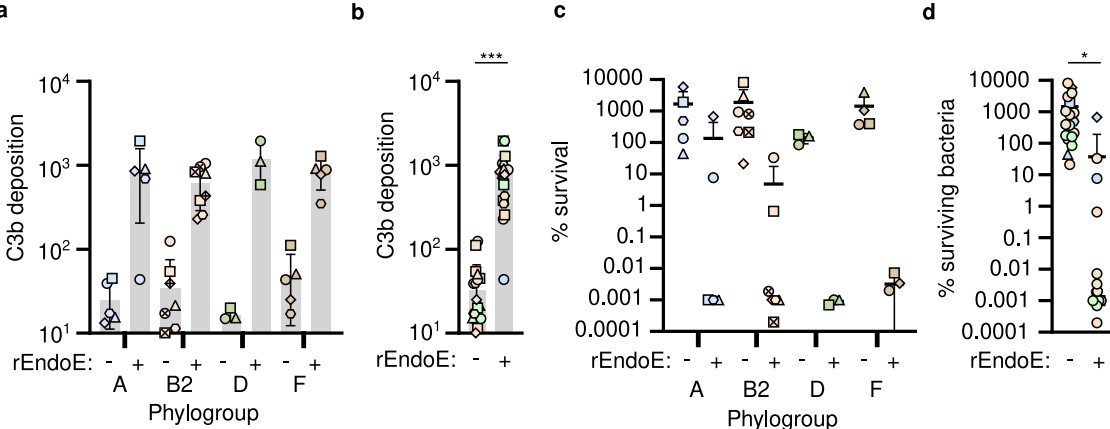

**Fig. 6 | Therapeutic targeting of capsule K1 enhances serum killing of *E. coli*.** **a**, **b** The effect of rEndoE capsule depolymerase on deposition of complement C3b on K1 encapsulated *E. coli* strains after incubation in human complement-replete serum. Each data point represents C3b deposition for one *E. coli* isolate, representing the mean from *n* = 2 independent experiments. In (**a**), each *E. coli* isolate is represented by a different symbol within the phylogroup. Mean ± s.d. are displayed. Total number of isolates in each phylogroup are *n* = 5 for A, *n* = 8 for B2, *n* = 3 for D and *n* = 5 for E. In (**b**), data points from **a** are pooled for all *E. coli* isolates (*n* = 21, with mean ± s.d. displayed). Each *E. coli* isolate is coloured by phylogroup (blue = phylogroup A, orange = phylogroup B, green = phylogroup C, brown = phylogroup D) and differentiated by symbol. Statistical significance tested for C3b deposition by paired two-sided Student's *t* test (rEndoE - vs. rEndoE+ ****p* < 0.0001). **c**, **d** The

effect of rEndoE capsule depolymerase on deposition of survival of K1 encapsulated *E. coli* strains during incubation in human complement-replete serum. Each data point represents % survival of one *E. coli* isolate, representing the mean from *n* = 2 independent experiments. In (**c**), each *E. coli* isolate is represented by a different symbol within the phylogroup. Mean ± s.d. are displayed. Total number of isolates in each phylogroup are *n* = 5 for A, *n* = 8 for B2, *n* = 3 for D and *n* = 5 for E. In (**d**), data points from **c** are pooled for all *E. coli* isolates (*n* = 21, with mean ± s.d. displayed). Each *E. coli* isolate is coloured by phylogroup (blue = phylogroup A, orange = phylogroup B, green = phylogroup C, brown = phylogroup D) and differentiated by symbol. Statistical significance tested for C3b deposition by paired two-sided Student's *t* test (rEndoE- vs. rEndoE+ **p* < 0.0101). Source data are provided as a Source Data file.

and the capsule has remained a distinctive feature of this successful lineage ever since. We recently established the relative invasiveness of *E. coli* lineages by comparing their colonisation capacity with their capacity to cause BSIs and found that CC95 is significantly more pathogenic than other *E. coli* lineages[81]. The success of the CC95 clone in causing human BSIs is likely to be in part driven by the capacity of the K1 capsule to subvert the innate immune response. Other virulence factors are likely to additionally contribute to the capacity of CC95 to cause invasive infections. Of note, 80.3% of K1 encapsulated CC95 isolates carried a PAI downstream of the K1-*cps* locus. This PAI was characterized by the presence of the *ireA* gene and *papGII* gene operon involved in iron-scavenging and bacterial adhesion. Of note, the *papGII* gene operon has been recently identified using a GWAS approach as one of the bacterial traits distinguishing invasive from non-invasive uropathogenic *E. coli*[60]. In addition, this PAI is located in isolates of other lineages and also carried other accessory genes that could benefit bacterial survival or infer virulence properties, including HlyA[63], CNF1[64], Sat[65], Sfa[66], Ila[67] and aerobactin[68]. Thus, more comprehensive analysis is required to elucidate the function of PAI-encoded virulence genes as molecular determinants of BSIs. This is suggestive that phenotypes encoded on the PAI provide mechanisms for *E. coli* to enhance their survival and virulence capacity to cause BSIs. The fact that capsule with the same polySia K1 chemical composition has evolved independently in virulent lineages of other bacterial pathogens, namely group B *Neisseria meningitidis*, *Moraxella nonliquefaciens* and *Mannheimia haemolytica* A2[82–84], further suggests that polySia enhances bacterial survival and/or virulence.

The 'origin' of the K1-*cps* encoding K1 encapsulation and 'transfer mechanism' remains unclear. Our data reveal multiple independent introductions of the K1-*cps* locus into *E. coli* populations. This can partially be attributed to horizontal gene transfer (HGT) of the capsule locus between *E. coli* populations. Downstream of the K1-*cps* locus, we observed the presence of a PAI with a non-uniform gene content truncating the *pheV* tRNA gene in all lineages. The gene synteny in this PAI was considered as evidence to infer which lineages acted as donors of the K1-*cps* locus. Furthermore, this PAI can form a circular excision

product that may be exported by larger MGEs[61]. The most likely transfer mechanisms of K1-*cps* are phage transduction and natural competence as the *cps* locus in *E. coli* does not possess an integrase or conjugation machinery.

Surprisingly, our analysis indicated that CC95 has not been the donor clone for most K1-*cps* locus transfers into recipient lineages, despite being the most dominant K1-positive *E. coli* clone and existing for over 250 years. Instead, our analysis indicated transfer of the K1-*cps* from phylogroup A to phylogroup B2 and F, and multiples transfers between lineages of phylogroup B2. The simplest hypothesis to explain why the dominant K1-*cps*+ clone (CC95) does not serve as the donor is the existence of evolutionary barriers preventing HGT. The capsule itself could provide a physical barrier that prevents infection of CC95 donors by phage that can lysogenize and infect the recipient[85]. There could be moments of capsule inactivation or swap that could be extremely deleterious for CC95 as it is a major virulence determinant, which could prevent the long-term persistence of other capsule types in CC95[86]. In addition, or alternatively, CC95 and recipient/donor *E. coli* could be rarely present in sufficient densities in the same environmental niche to permit HGT. This seems unlikely given the global success of the CC95 clone, unless CC95 effectively suppresses most of *E. coli* competitors in the environment. A recent large-scale analysis showed that genes encoding group 2 capsules were detectable in 11 genera within the Enterobacteriaceae belonging to 3 bacterial families[87], highlighting that transfer of *cps* loci among species. In support of this, searching the NCBI genome catalogue detected the genes of the K1-*cps* locus in *Enterobacter hormaechei*[88] and *Shigella flexneri* (GenBank: CP041620). Additional studies to understand HGT mechanisms of *cps* transfer, and the barriers to transfer, are required.

Our analysis demonstrates that the K1-*cps* can be efficiently acquired by *E. coli*, and that K1 capsule expression provides protection against the human immune system. If K1-*cps* acquisition is so beneficial to *E. coli*, then the fact that 75% of all BSI *E. coli* isolates do not possess K1-*cps* genes suggests that there are costs to a bacterium associated with K1 capsule expression. The immediate costs of K1 capsule production are reduced growth rates, as the production of K1

polySia imposes a substantial metabolic burden[89], and an increased susceptibility to killing by bacteriophage that targets the K1 capsule[90]. In addition, capsule synthesis may cause a drastic reductions in plasmid conjugation[86], which could limit the acquisition of new traits required to maintain survival or competitiveness in the local environment, such as antimicrobial resistance genes which are usually disseminated through plasmid conjugation. The fact that K1 capsule expression is thermoregulated[26,91], downregulated in the urine/bladder environment[92] and switched off in the brain environment[40], is further suggestive that there are costs associated with K1 expression inside and outside of the mammalian host. These findings highlight the need for future studies that establish the dynamics and functions of K1 capsule expression on the pathway to causing infection.

Much attention and research effort has been directed towards the increasing levels of antimicrobial resistance in *E. coli*[75]. In particular, the rise of MDR isolates belonging to CC131 lineage has been a global concern[93]. The dominant K1 positive lineage is CC95. This lineage has remained a dominant cause of BSIs for over two decades[50,51,94,95] and a major cause of UTIs[51,60,96]. The dominance of this lineage as a cause of invasive infections is despite CC95 classically being a non-MDR lineage, with the population displaying susceptibility to third generation cephalosporins and carbapenems but strain-dependent resistance to many other antibiotics. Of concern, is the recent report of CC95 isolates in which MDR and K1 capsule expression have converged[97]. As the administration of antibiotics to treat a bacterial infection disturbs the normal gastrointestinal microbiota[98], can select for colonisation with pathogens[99], and can increase the risk of infectious and non-infectious diseases[100,101], there is a strong incentive for the development of antibacterial therapies that target specific pathogens without disrupting the microbiota. Here, we have demonstrated that therapeutic targeting of the K1 capsule using a bacteriophage-derived enzyme can sensitize K1 expressing *E. coli* isolates from distinct genetic backgrounds to the human immune response. This K1-capsule depolymerase effectively prevents and treats invasive *E. coli* infections in experimental animal models[45,46], and similar therapeutic approaches have been effective in enhancing protective immunity against some of the most clinically important bacterial pathogens[102–105]. As our data indicates that around ~25% of *E. coli* causing BSIs are K1 encapsulated, such a therapeutic approach would have a considerable potential to effectively control invasive *E. coli* infections whilst significantly lowering antibiotic usage and its adverse effects.

## Methods

### Ethics
Human serum was prepared from blood obtained from healthy donors, approved by the Regional Ethics Committee and Imperial College Healthcare NHS Trust Tissue Bank (Regional Ethics Committee approval no. 17/WA/0161, Imperial College Healthcare Tissue Bank Human Tissue Authority license no. 12275, and Imperial College Research Ethics Committee no. 19IC5166). All samples were collected after receiving signed informed consent from all participants.

### Bacterial strains
To delete the *neuC* gene present in different *E. coli* lineages, we made use of λ-red recombineering[106] or the thermosensitive allelic replacement using the plasmid pKO3blue[107]. The mutants generated using λ-red recombineering were generated using primers listed in Dataset S5. For mutants generated using thermosensitive allelic replacement, the desired *neuC* chromosomal flanking regions were amplified by PCR using the primers listed in Dataset S5. Then, a fusion PCR from both flanking regions was performed and cloned into [107]pKO3blue, using enzymatic digestion and ligation following standard protocol. The generated plasmids are listed on Dataset S5. To perform the allelic replacement, the desired pKO3blue plasmid was introduced into the correspond strain and the transformants were selected on LB agar

plates supplemented with 20 µg/ml chloramphenicol and incubated at 32 °C. Then, one colony was grown at non-permissive temperature, 42 °C, to force the plasmid to integrate into the chromosome and plated on LB agar containing 20 µg/ml chloramphenicol and X-gal (5-bromo-4-chloro- 3-indolyl-B-D-galactopyranoside). Light blue colonies, which are indicative of plasmid integration, were grown in LB broth at 32 °C. Then, ten-fold serial dilution of the overnight cultures was plated on LB agar containing X-gal and sucrose 5%, which favour plasmid loss. Plates were incubated at 32 °C for 24 h and white colonies, which are indicative that the plasmid is loss, were screened for chloramphenicol sensitivity. A PCR was performed to differentiate wild-type and mutant strains and mutants were verified by sequencing. Isogenic strains constructed and used are shown in Dataset S6. Strains used in population-based screens are shown in Dataset S7. Strains used in rEndoE-based assays are shown in Dataset S8.

### Expression of GFP-endoNA2 (rEndoNA2-GFP) and measurement of K1 capsule expression by *E. coli*
An *E. coli* strain carrying the pFEndoNA2 plasmid was previously generated[108] and accessed through AddGene. The pFEndoNA2 plasmid expresses the GFP-EndoNA2 fusion protein (rEndoNA2-GFP) with a C-terminal 6xHis tag, that includes the catalytically inactive form of the endosialidase from bacteriophage PK1A2. rEndoNA2-GFP was expressed by culture of *E. coli* in LB media at 37 °C with shaking (180 rpm) until absorbance at $OD_{600}$ reached 0.6. After addition of 1 mM IPTG, *E. coli* were cultured for a subsequent 24 h at 30 °C with shaking (180 rpm). Bacteria were lysed, and rEndoNA2-GFP was purified using a HIStrap Nickel column (GE Healthcare Life Sciences) and affinity chromatography (ÄKTA Pure, GE Healthcare Life Sciences). Purified rEndoNA2-GFP was dialysed against PBS at 4 °C.

Levels of K1 capsule expression were quantified by measuring rEndoNA2-GFP binding to *E. coli* using flow cytometry. $6 \times 10^6$ of logarithmic phase bacteria were incubated on ice for 1 h with 10 µg/ml rEndoNA2-GFP. Bacteria were washed in PBS before fixing in 1% (w/v) PFA. Fluorescence was measured by flow cytometry analysis using an Amnis CellStream Benchtop Flow Cytometer. Data was collected using CellStream Acquisition and Analysis Software and analysed using FlowJo v10.8. Gating strategy is shown in Supplementary Fig. 4c.

### Expression of rEndoE
The pQE30.EndoE plasmid[45] was used to express the EndoE protein with a C-terminal 6xHis tag, that includes the catalytically active form of the endosialidase from phage K1E. EndoE was expressed by culture of *E. coli* in LB media at 37 °C with shaking (180 rpm) until absorbance at $OD_{600}$ reached 0.35. After addition of 0.5 mM IPTG, *E. coli* were cultured for a subsequent 18 h at 30 °C with shaking (180 rpm). Bacteria were lysed, and EndoE was purified using a HIStrap Nickel column (GE Healthcare Life Sciences) and affinity chromatography (ÄKTA Pure, GE Healthcare Life Sciences).

### Flow cytometry measurement of complement C3b deposition on the *E. coli* surface
Levels of C3b deposition on *E. coli* following exposure to human serum were measured by flow cytometry. $6 \times 10^6$ of logarithmic phase bacteria were incubated at 37 °C for 30 min in PBS, 3% pooled active human serum or 3% pooled heat-inactivated human serum. Bacteria were washed twice in PBS, followed by centrifugation at $2058 \times g$ for 5 min and removal of the supernatant. C3b deposition on the bacterial surface was detected by incubation on ice for 30 min with polyclonal FITC-conjugated goat anti-human-C3b (dilution 1:2000) (MP Biomedical). Bacteria were washed as previously described before fixing in 1% (w/v) PFA. Fluorescence was measured by flow cytometry analysis using an Amnis CellStream Benchtop Flow Cytometer. Data was collected using CellStream Acquisition and Analysis Software and analysed using FlowJo v10.8.

## Susceptibility of *E. coli* to human serum

Bacteria were grown to logarithmic phase and washed twice with gelatin-Veronal-buffered saline plus magnesium and calcium ions (pH 7.35) (GVB++; Sigma Aldrich). Pooled human serum was diluted 1:3 in GVB++ and prewarmed to 37 °C. Bacteria were resuspended in GVB++ and mixed with serum solutions 1:2 to give a final concentration of ~10⁷ CFU/ml in a total volume of 125 μl containing 22% serum. *E. coli* and serum mixtures were incubated at 37 °C for 3 h, and the number of surviving bacteria were quantified by serial dilution and incubation on LB agar overnight. Prewarmed, heat-inactivated (56 °C for 30 min) serum served as a control.

## Bioinformatics

**In silico detection of the K1 capsule locus**. The BSAC, NORM and phenotypically confirmed K1+ collection were screened using an in silico PCR of the assemblies for each of the six *neu* genes. If at least one *neu* gene was detected the isolates were further investigated as putative K1 loci. The K1 *neu* gene alleles from the NORM and BSAC genomes were used to screen 12230 genomes in the Horesh collection, that contains the historical isolates published by Baker et al., using the Horesh et al. BIGSI database to identify further K1 genomes[52]. The putative K1 genomes were screened using an in silico PCR of the assemblies for the entire *kpsF-kpsM* locus. A full length K1 loci was used as a reference database for ARIBA to assemble the putative K1 loci for the genomes which the locus PCR of the assemblies did not return a product. Alleles were assigned for each locus that assembled into a single contig, and panaroo K1 loci gene groups using any difference in the sequence to define a new allele, partial alleles (<95% length) were named as such, only genomes with 14 full length K1 gene sequences or a confirmed K1 phenotype were considered K1. The DNA alleles were translated using EMBOSS transeq and assigned protein alleles.

## Species phylogenetic tree

The entire NORM[50] and BSAC collection[51] (K1 and non-K1), the geno-typically confirmed K1 from Horesh et al.[52] and the phenotypically confirmed K1+ collection presented in this work (accession list in Dataset S1), were mapped against *E. coli* strain EC958 (accession HG941718; https://www.ncbi.nlm.nih.gov/nuccore/HG941718.1) to create a species-wide phylogeny using iqtree (version 2.2.0.3) with the fast and GTR settings[109]. Phylotypes were assigned using Clermont-Typer (version 0.6.3)[110] and corrected based on the phylogenetic clusters, and STs were assigned by mlst https://github.com/tseemann/mlst (version 2.9) which makes use of the PubMLST database[111], and overlaid on the phylogeny in Microreact (version 2.1.8)[112].

## Lineage analysis

To investigate the age of K1 isolates, the NORM collection isolates belonging to the CC59, and CC95, CC131 and CC10 lineages as defined in ref. 50 were dated using BactDating[113]. Briefly, isolates were mapped to a lineage reference, and gubbins (version 2.4.1)[114] used to remove recombination to generate the BactDating (version 1.1.0) input. The lineages were run in triplicate for multiple bactdating models with MCMC chains of 100,000,000, convergence was assessed using the R (version 4.0) coda package (version 0.19) considering the effective sample size (>200) and Gelman-Rubin diagnostic (-1). The final model was compared to a randomised dates model considering the deviance information criterion.

For each K1-*cps*+ lineage, we obtained a complete reference genome by conducting long-read sequencing and generating a hybrid assembly. This allowed us to confidently characterize the gene synteny present downstream of the K1-*cps* locus. The long-read sequencing and subsequent hybrid assembly was performed as previously described in Arredondo-Alonso et al.[115] to obtain a complete genome sequence from each K1+ lineage present in the NORM collection. We selected the following genomes 30348_1#322 (CC141), 30224_1#187 (CC80),

30224_1#150 (CC144), 30134_6#117 (CC95), 30345_1#279 (CC131), 30348_2#141 (CC1193), 30224_1#376 (CC62), 30348_2#375 (CC10) and 30348_2#365 (CC59). For each genome, we rotated their starting position considering the *dnaA* gene using circlator (fixstart, version 1.5.5) and extracted with bedtools (−getfasta) 50 kbp upstream *kpsM* and 50 kbp downstream the integrase *intB* gene found in the island adjacent to the K1 locus. We annotated the extracted regions with prokka (−genus Escherichia,−species coli) and visualized their gene synteny similarity with clinker (default settings, version 0.0.21).

## K1-*cps* synteny in *E. coli* near-complete genomes

We retrieved all the complete genomes from the RefSeq database (queried in October 2022) annotated as ST95 in Enterobase[116]. The starting position of the chromosomal sequence was rotated considering the *dnaA* gene using circlator (fixstart, version 1.5.5)[117] The genomes were annotated with prokka (version 1.14.6) considering the *Escherichia* scheme (−genus Escherichia,−species coli)[118]. The pangenome of those ST95 complete genomes was computed with Panaroo (version 1.2.3)[119] in the strict mode (−clean-mode,−no_clean_edges) considering the gff files computed by prokka as input. The module 'panaroo-gene-neighbourhood' was used to group the genomes based on the gene graph neighbourhood of the 10 genes (−expand_no 10) adjacent (upstream and downstream) of the *kpsM* and *kpsF* genes. We focused our analyses on the synteny found by panaroo downstream of the *kpsF* gene and further identified the presence of the pathogenicity island (PAI-V) characterized by the insertion of a ~50 kbp region in the *pheV* tRNA gene mediated by a P4-like integrase (*intB* gene) with similarity to the RefSeq amino acid sequence EG12364. The size of the PAI was determined by extracting the size of the genome block surrounding the truncated copies of the *pheV* gene. To visualize the most predominant synteny blocks computed by panaroo, we considered the complete chromosomal sequences from the RefSeq assemblies GCA_002285755.1 and GCA_001280405.1 and extracted the nucleotide sequence corresponding to 50 kbp upstream and 70 kbp downstream of the K1-*cps* locus with bedtools (−getfasta, version 2.30.0)[120]. The extracted regions were annotated with prokka (−genus Escherichia,−species coli) and the visualization of their gene clusters similarity was performed with clinker (default settings, version 0.0.21)[121].

We determined the presence of this PAI in the NORM CC95 isolates by inspecting the tRNA *pheV* gene in the short-read assemblies of the NORM collection using a customized database in abricate (version 1.0.1, https://github.com/tseemann/abricate). The presence of a truncated copy of the *pheV* gene in the same short-read contig as for the *kpsF* gene indicated the possible presence of the PAI. We confirmed this by nucleotide searching (blastn) for the PAI-related integrase *intB* gene on the same contig as the other truncated copy of *pheV*. Furthermore, we determined the size of the PAI in the cases where the two copies of the truncated *pheV* gene were present on the same short-read contig. Lastly, the presence on the same short-read contig of an intact *pheV* gene downstream the *kpsF* was considered as evidence for the absence of the PAI. The R library ggtree (version 3.5.1.902)[122] was used to plot the presence/absence of the PAI in the NORM CC95 phylogeny. To screen the virulence gene content present in the CC95 PAI and in the PAIs associated to each of the hybrid assemblies generated from the other K1-*cps*+ lineages (Dataset S4), we used amrfinder (version 3.10.18) with the flags −plus and −organism (indicating *Escherichia* as the taxonomic group)[123].

## SNP profile of the region 2 of the K1-*cps* locus

To investigate and support the introductions of the K1-*cps* locus, we used parsnp (version 1.2)[124] to compute a pairwise matrix with the SNP differences considering as input a concatenation of the alignments produced by panaroo on the genes corresponding to the region 2 of the locus. Isolates were grouped according to their CC lineage, and the median number of SNPs found within and between CC lineages

reported. To visualize isolates with a similar SNP profile, we embedded the SNP distances computed by parsnp into two dimensions using the t-Distributed Stochastic Neighbour Embedding algorithm as implemented in the Rtsne package (version 0.15) (perplexity = 50, theta = 0.5, is_distance = TRUE)[125].

## Statistics and reproducibility

All quantitative data were analysed and graphed using GraphPad Prism 8.4.3 software. All data are displayed as mean ± s.d. calculated using the GraphPad Prism 8.4.3 software, unless indicated otherwise. Statistical details of the experiments are provided in the respective figure legends and in each methods section pertaining to the specific technique applied. Bactdating (version 1.10) was used to date the phylogenies of the lineages CC59, CC95, CC131 and CC10. The bactdating models were run in triplicate with MCMC chains of 100,000,000, their convergence was assessed using the R package coda (version 0.19) considering the effective sample size of the parameters (>200) and the Gelman-Rubin diagnostic (-1). The final model was compared to a randomised dates model using bactdating (version 1.10). No statistical method was used to predetermine sample size. No data were excluded from the analyses. The experiments were not randomized. The investigators were not blinded to allocation during experiments and outcome assessment.

## Reporting summary

Further information on research design is available in the Nature Portfolio Reporting Summary linked to this article.

## Data availability

The Illumina sequencing data generated in this study have been deposited in the NCBI database under accession codes listed in Dataset S1. The publicly available Illumina sequencing data used in this study have previously been deposited in the NCBI database under accession codes listed in Dataset S1. The accessions of the complete genomes from CC95 retrieved from RefSeq are listed in Dataset S3. The data generated in this study are provided in the Supplementary Information or Source Data file or on Microreact https://microreact.org/project/cm8f8adPyWoQqemAAgvrmi-k1-context. The Oxford Nanopore Technologies sequencing data used to generate (near-)complete chromosomal sequences for: 30348_1#322 (CC141), 30224_1#187 (CC80), 30224_1#150 (CC144), 30134_6#117 (CC95), 30345_1#279 (CC131), 30348_2#141 (CC1193), 30224_1#376 (CC62), 30348_2#375 (CC10) and 30348_2#365 (CC59) are available through the accessions listed in Dataset S4, and their complete chromosomal sequences shared in the figshare project https://doi.org/10.6084/m9.figshare.21610674[126]. Source data are provided with this paper.

## Code availability

An Rmarkdown document with the code and files required to generate the figures and results presented here is available at https://gitlab.com/sirarredondo/k1_manuscript.

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

## Acknowledgements

We thank David A.B. Dance (University of Oxford) for assistance in collating the strain collection. We thank the Norwegian laboratories for establishing the NORM collection through the Norwegian Surveillance System on Resistant Microbes. This work was financed by the Medical Research Council (MRC, UK) through grants MR/K018396/1, MR/N012542/1 and MR/R009937/1 to P.W.T.; the Biotechnology and Biological Sciences Research Council (BBSRC, UK) grant BB/V006495/1 to A.J.M.; the Wellcome Trust grant 206194 to N.R.T., by the European Union Horizon 2020 research and innovation programme under the Marie Skłodowska-Curie Actions (grant No. 801,133 to S.A.-A. and A.K.P.). This work received funding from the Trond Mohn Foundation (grant identifier TMS2019TMT04 to A.K.P., R.A.G., Ø.S., P.J.J., and J.C.). This work was partially supported by the European Research Council (grant No. 742,158 to J.C.), Joint Programming Initiative in Antimicrobial Resistance (JPIAMR Third call, STARCS, JPIAMR2016-AC16/00039) to S.A.-A., and Engineering and Physical Sciences Research Council (EPSRC, UK) grant EP/X026671/1 to J.R.P.

## Author contributions

These authors contributed equally: George Blundell-Hunter, Zuyi Fu, Rebecca A. Gladstone. S.A.-A. and R.A.G. performed the bioinformatic, genomic analyses and worked on the manuscript writing. E.C.-G., P.J.J., Ø.S., A.K.P., F.C., S.C.-B., M.A.C., M.A.A., M.V., A.C., C.H., N.K., A.M., and J.N.R. sampled isolates. G.B.-H., Z.F., A.F.-S., J.L., and A.J.M. conducted the experimental work. A.J.M., P.W.T., Ø.S., P.J.J., J.C., J.R.P., and N.R.T. acquired the funding. S.A.-A., A.J.M., P.W.T., and J.C. designed the study and contributed to the analyses. S.A.-A., A.J.M., and J.C. wrote the original draft of the manuscript and revised the manuscript with input from all authors. All authors read and approved the final paper.

## Competing interests

The authors declare no competing interests.

## Additional information

[1]Department of Biostatistics, University of Oslo, 0317 Oslo, Norway. [2]Parasites and Microbes, Wellcome Sanger Institute, Cambridge, UK. [3]School of Pharmacy, University College London, London, UK. [4]Department of Infectious Disease, Centre for Bacterial Resistance Biology, Imperial College London, London, UK. [5]Great Ormond Street Institute of Child Health, University College London, London, UK. [6]Department of Pharmacy, Faculty of Health Sciences, UiT The Arctic University of Norway, Tromsø, Norway. [7]Norwegian National Advisory Unit on Detection of Antimicrobial Resistance, Department of Microbiology and Infection Control, University Hospital of North Norway, Tromsø, Norway. [8]University of Missouri Kansas City, Kansas City, USA. [9]Division of Infectious Diseases, Children's Mercy Hospital Kansas City, UMKC School of Medicine, Kansas City, USA. [10]Unidad de Investigación Médica en Enfermedades Infecciosas y Parasitarias, Hospital de Pediatría, Centro Médico Nacional Siglo XXI Instituto Mexicano del Seguro Social, Mexico City, Mexico. [11]Facultad de Medicina, Benemérita Universidad Autónoma de Puebla, Puebla, Mexico. [12]Lao-Oxford-Mahosot Hospital-Wellcome Trust Research Unit (LOMWRU), Microbiology Laboratory, Mahosot Hospital, Vientiane, Lao PDR. [13]Faculty of Medicine, Chair of Microbiology, Jagiellonian University Medical College, Czysta str. 18, 31-121, Kraków, Poland. [14]British Society for Antimicrobial Chemotherapy, Birmingham, UK. [15]Institute of Microbiology and Infection, College of Medical and Dental Sciences, University of Birmingham, Birmingham, UK. [16]Laboratory of Pathology and Molecular Biology (LPBM), Gonçalo Moniz Research Institute, Oswaldo Cruz Foundation, Salvador, Brazil. [17]Faculdade de Farmácia, Universidade Federal da Bahia, Salvador, Brazil. [18]London School of Hygiene and Tropical Medicine, London, UK. [19]Helsinki Institute of Information Technology, Department of Mathematics and Statistics, University of Helsinki, Helsinki, Finland. [20]These authors contributed equally: George Blundell-Hunter, Zuyi Fu, Rebecca A. Gladstone. ✉e-mail: jukka.corander@medisin.uio.no; peter.taylor@ucl.ac.uk; a.mccarthy@imperial.ac.uk

