## [Peer review file · Nature Communications]

REVIEWER COMMENTS

Reviewer #1 (Remarks to the Author):

In their manuscript entitled " Evolutionary and Functional History of the Escherichia coli K1 Capsule" by Sergio Arredondo-Alonso et al study the spread of capsule K1 in E. coli and its expression. Using phylogenies they date the multiple acquisitions of this island across Sequence types and show that it is well expressed across backgrounds and provides a benefit against serum. This suggests a potential success of a vaccinal approach against K1 capsid that is an important virulence factor.

It is an interesting study, it is well done with the proper methods and I could not find any other thorough analysis of K1 with genomics.

As such, I have only minor comments.

1) Multiple acquisition of the K1 are suggested throughout the text but no figure is supporting it. Could a phylogeny of the kps operon help resolve that, showing among other things the different STs with different colors. That phylogeny could support more clearly the scenarios relative to the propagation of K1 within the species. (Figure S2 is doing part of the job but requires a lot of thinking...)

2) In Figure 1, to infer selection of the K1 at a broader scale, it could be interesting to have also the other capsular types mapped on the same phylogeny.

3) The PAI downstream from kpsF in Figure 2 is showing a very nice signal of acquisition. Yet, multiple loss and acquisition are also evident on the phylogeny. Are there any associated phenotypes in the isolation of these clones.

4) While benefits of K1 are proposed there is no discussion of the associated costs.

5) One of the underlying questions that drives the present work, is whether the capsule would be properly expressed across background; There could be more arguments about why we do not expect it.

I really appreciated the discussion that is humble and well balanced. Overall it is a very solid study.

Olivier Tenaillon (I always sign my reviews)

Reviewer #2 (Remarks to the Author):

The manuscript titled “Evolutionary and Functional History of the Escherichia coli K1 Capsule” by Arredondo-Alonso et al. describes the role and evolution of the K1 capsule in infection by E. coli from extra intestinal sources. The authors rely on the reanalysis of ExPEC genomes from two large studies that have been deposited in the public databases. The work is focused on the presence and absence of the K1 capsule loci in the ExPEC isolates, but does not examine the prevalence in any of the diarrheagenic E. coli, which are also a significant source of blood stream infections, and may actually be the more common source, but are totally ignored in this study. The phylogenomic studies lack significant details which would allow a reader to be able to recreate the analyses presented. Evolutionary studies attempt to identify when the kps locus was acquired in these specific genomic backgrounds. These studies lack rigour and should be completed using statistics to support the conclusions. The authors spend a significant amount of time on the pathogenicity island associated with the kps loci, yet spend very little time describing what is within this actual PAI. Functional analyses indicated that the capsule was functional and could bind the ligand and prevent killing by complement. Overall, there are significant biases in the selection of the isolates in this study, there is a lack of statistical tests included in the analyses and the data presentation especially in the supplemental data is lacking resulting in a lack of enthusiasm for this study and lack of confidence in the conclusions.

Major comments

The K1 was only considered in the “most predominant extraintestinal lineages” which will result in a bias to previous and possibly more ancestral introductions. For example, the authors suggest that the CC95 isolates are the most successful clones, but the selection of the isolates is biased by the inclusion and examination of the genomes. These biases result in the conclusions of this study being significantly flawed and these flaws are not addressed in a meaningful way in the current study. The authors may suggest that they have not generated the data thus are not responsible for the biases, but they do have the ability to select any of the >50000 E. coli genomes that are currently in the public domain to address these biases.

Only ¼ of the BSI contains the K1 capsule – the authors have not demonstrated that this is clinically or statically significant. This is a significant gap in the importance of the K1 capsule in this group of pathogens. This is significant considering that the vast majority of these isolates are from blood. Additionally the date of 250 years ago indicated in line 190 is not supported by the Figure 2, as this figure is for the presence or absence of the PAI associated with the kps cluster, not the actual cluster itself. Other dates of 500 years ago are mentioned in the discussion, but not in the results. These dates are not well supported in the results, and the lack of statical methods included with these analyses makes these dates not well supported.

The lack of any statistical tests associated with the dating in Figures 2 and 3 makes any conclusions made from this data suspect. Additionally, Figure 3 as presented is weak and unsupported and not well described in the results.

The isolate IDs provided in Supplementary Table 1 (assumed to be) does not match the isolate numbers provided in the ENA database for the provided accession numbers. The authors should ensure that these designations are consistent.

The authors mention increases in fitness in the abstract, introduction and discussion but have no actual measures of fitness. Mentions of fitness should be removed unless the authors want to complete these studies correctly. The same can be said for increasing antimicrobial resistance and invasiveness – none of these phenotypes are actually measured.

A general lack of labeling of any of the supplemental documents makes the supplemental information difficult to navigate and essentially useless. An experienced group such as is assembled here has not actually seen and reviewed this manuscript any detail, is asleep at the wheel or does not care enough to even look at these documents for clarity. Any of these options leads this reviewer to suggest that the remainder of the manuscript is suspect in terms of attention to detail. This may have something to do with the 25 authors in the author list, but only 11 authors have a listed contribution other than manuscript revision/ If the role of 14 authors was ONLY to review and revise the manuscript, these issues should not be present. The corresponding authors should review the rules for authorship to ensure that each of these named individuals actually meet the established criteria to be an author.

Minor comments

The authors state that 8 genes are conserved and 6 are not in this gene cluster, but in Figure 1 it is not clear which are which – this should be clarified somewhere in the manuscript either in the figure legend or the text.

Line 52 – define “pandemic lineages” this term has different meanings in today’s terms compared to previous eras.

The authors should define that UTI are invasive infections. Many believe that these infections are not invasive.

Define “enhances E. coli fitness”

In Supplemental Table 1 what is the difference between UTI and Urine?

POINT-BY-POINT RESPONSE

Reviewer #1

In their manuscript entitled " Evolutionary and Functional History of the Escherichia coli K1 Capsule" by Sergio Arredondo-Alonso et al study the spread of capsule K1 in E. coli and its expression. Using phylogenies they date the multiple acquisitions of this island across Sequence types and show that it is well expressed across backgrounds and provides a benefit against serum. This suggests a potential success of a vaccinal approach against K1 capsid that is an important virulence factor. It is an interesting study, it is well done with the proper methods and I could not find any other thorough analysis of K1 with genomics.

We thank the reviewer for their interest in the study and their comments.

Minor comments

1. Multiple acquisition of the K1 are suggested throughout the text but no figure is supporting it. Could a phylogeny of the *kps* operon help resolve that, showing among other things the different STs with different colors. That phylogeny could support more clearly the scenarios relative to the propagation of K1 within the species. (Figure S2 is doing part of the job but requires a lot of thinking...)

We thank the reviewer for this suggestion. We agree that Figure S2 provided a quantitative assessment of the SNP differences supporting the propagation of K1 but lacked a visual and intuitive visualization to support our interpretation in the manuscript.

Following the reviewer's suggestion, we produced a phylogeny with *iqtree* considering as input the SNP alignment of the K1-*cps* region 2 and a GTR model. However, the level of the resolution of the resulting phylogeny was not sufficient to show the distinct K1-*cps* propagation events. The generated figure is available at the link below together with the code used to generate and compile the figure.

Figure: https://gitlab.com/sirarredondo/k1_manuscript/-/raw/main/K1_code_files/figure-gfm/unnamed-chunk-10-2.png

Code:

https://gitlab.com/sirarredondo/k1_manuscript/-/blob/main/K1_code.md#snp-differences-in-the-region-2-of-the-k1-cps-locus.

As an alternative, and motivated by the reviewer's comment, we used T-distributed stochastic neighbour embedding (t-SNE) to embed the SNP matrix into 2 dimensions and colour each point (isolate) with its corresponding ST. The embedding of high-dimensional data into a low-dimensional representation for visualization purposes has been previously adopted for the exploration of the bacterial population structure and gene content (Abudahab et al. 2018 DOI: 10.1099/mgen.0.000220, Lees et al. 2022 DOI:10.1098/rstb.2021.0237). This analysis shows the presence of distinct clusters in which points from distinct STs indicated a similar SNP profile. This new figure further supports our conclusions in the manuscript that there have been multiple K1 propagation events. To include this figure, we have split the current Figure S2 into a multipanel figure for which Figure S2B corresponds to this t-SNE embedding based on the SNP profile of the K1 isolates.

2. In Figure 1, to infer selection of the K1 at a broader scale, it could be interesting to have also the other capsular types mapped on the same phylogeny.

We thank the reviewer for this interesting suggestion. However, due to the lack of any publicly available tool designed and validated to type *E. coli* capsules from whole-genomic sequencing data, we cannot perform a robust and sensitive exploration of capsule variation in the non-K1 *E. coli* population that could permit the mapping of other capsular types onto the phylogeny. We believe that the design of a capsular typing tool for *E. coli* is an interesting aspect worth investigating in further research.

3. The PAI downstream from *kpsF* in Figure 2 is showing a very nice signal of acquisition. Yet, multiple loss and acquisition are also evident on the phylogeny. Are there any associated phenotypes in the isolation of these clones.

The reviewer correctly points out that our analysis reveals several instances of loss and acquisition of the PAI spread in the phylogeny of CC95 (Figure 2A). This finding is consistent with a recent population analysis that reported gain, loss, and/or rearrangements of this PAI in UPEC isolates belonging to CC95, CC69, CC73 and CC131 lineages (Whitfield *et al.* 2006 DOI:10.1046/j.1365-2958.1999.01276.x; Biggel *et al.* 2020 DOI: 10.1038/s41467-020-19714-9). For Figure 2A, all the CC95 genomes belonged to the NORM collection which are isolates derived from bloodstream infections thus the presence/absence of the PAI cannot be associated to a distinct phenotype. To understand whether the PAI is associated with BSI isolates or UPEC, we interrogated the 44 RefSeq complete genomes that we employed to discover the PAI present downstream of the K1-cps locus. The metadata regarding the isolation source and its associated host has now been added as part of the old Table S2 (now Table S3). We could not observe any trend regarding the isolation source and the presence/absence of the PAI. This observation is now included in the manuscript.

Lines 223-227

Downstream of the locus, two predominant genome configurations were defined across CC95 i) the insertion of a pathogenicity island (PAI) that has resulted in the truncation of the tRNA *pheV* gene (26/44, 59%), and ii) the absence of the PAI and an intact *pheV* gene (16/44, 36%), while 2 genomes (2/44, 5%) showed each one a distinct gene synteny as determined by Panaroo.

Lines 236-238

We did not observe any trend in the association between the presence/absence of the PAI and the isolation source (BSI vs UPEC) of these 44 complete genomes (Table S3), indicating that acquisition or loss of this PAI is not exclusively linked to either type of invasive infection.

As PAIs have a mosaic structure and flexible accessory gene pool (Desvaux *et al.* 2020 DOI: 10.3389/fmicb.2020.02065), it could be possible that certain virulence genes are associated with specific phenotypes within CC95 and/or other lineages. Indeed, a recent genome-wide associated study identified the PAI-encoded P fimbriae-encoding *papGII* locus, encoding an adhesin, as the key feature that distinguishes invasive UPEC from non-invasive UPEC (Biggel *et al.* 2020 DOI: 10.1038/s41467-020-19714-9). Genetic analysis of the PAI in the CC95 lineage revealed a mosaic structure. By comparing the virulence genes of the isolates that lost the PAI with the closest isolate in the phylogeny having the PAI, we observed that the main difference resided in the absence of *papGII*

locus and an iron-acquisition (*ireA*) locus. No additional virulence genes were identified in the PAI of CC95 isolates.

Motivated by the comments of both reviewers, we further elucidated the virulence gene content of the PAI found in non-CC95 isolates located downstream the K1-*cps* locus using the amrfinderplus database. The analyses indicated that the virulence gene content found in the PAIs downstream of the K1-*cps* locus is highly dynamic with multiple gene losses/acquisitions. Notably, several characterized virulence factors were identified, as outlined in the text. Whilst it is tempting to speculate that these virulence factors play an important role in the onset of BSI, we believe a more comprehensive analysis - outside the scope of this manuscript - is required to elucidate the function of PAI-encoded virulence genes as molecular determinants of BSIs. We have updated the description of this analysis and the virulence gene content in the manuscript.

Lines 234-236

Because PAIs carry one or more accessory genes that encode virulence factors that often function as adhesins, iron-acquisition systems, host defence mechanisms or toxins ⁵⁹, the acquisition and loss of this PAIs is likely to impact phenotypes and infectivity.

Lines 239-247

The mosaic structure and flexible pool of virulence genes carried by PAIs could mean that a certain virulence gene is associated with the PAI in the CC95 lineage. However, a comparison of the virulence genes of the isolates that lost the PAI with the closest PAI+ isolate in the phylogeny indicated the main difference resided in the presence and absence of *papGII* locus encoding the P fimbriae and an iron-acquisition (*ireA*) locus. Interestingly, these genes have been recently identified as key features distinguishing invasive from non-invasive UPEC ⁶⁰. Additional genes that have roles in promoting bacterial survival or virulence were not identified on the PAI in CC95 isolates.

Lines 314-324

Of note, isolates from lineages in phylogroup A (CC10), phylogroup B2 (CC80, CC131, CC141, CC144, CC1193) and phylogroup F (CC59 and CC62) also frequently had a PAI inserted downstream of the K1-*cps* locus (Fig. 4). Despite high frequency in the PAIs from CC95, the gene encoding IreA was only additionally present in PAI from CC131, and the gene encoding PapGII was only present in the PAI from lineage CC141. Analysis of gene content of the PAI from isolates of other lineages demonstrates a heterogenous pool of virulence genes. This includes genes encoding characterized virulence factors α -hemolysin toxin (HlyA) ⁶³, cytotoxic necrotizing factor (CNF1) ⁶⁴, secreted autotransporter toxin (Sat) ⁶⁵, S-fimbrial adhesin (Sfa) ⁶⁶, the IrgA homolog adhesin (IIa) homolog adhesin ⁶⁷ and aerobactin ⁶⁸.

More comprehensive analysis is required to elucidate the function of PAI-encoded virulence genes as molecular determinants of BSIs.

Lines 432-440

This PAI was characterized by the presence of the *ireA* gene and *papGII* gene operon involved in iron-scavenging and bacterial adhesion. Of note, the *papGII* gene operon has been recently identified using a GWAS approach as one of the bacterial traits distinguishing invasive from non-invasive uropathogenic *E. coli*⁶⁰. Additionally, this PAI located in isolates of other lineages also carried other accessory genes that could benefit bacterial survival or infer virulence properties, including HlyA⁶³, CNF1⁶⁴, Sat⁶⁵, Sfa⁶⁶, Ila⁶⁷ and aerobactin⁶⁸. Thus, more comprehensive analysis is required to elucidate the function of PAI-encoded virulence genes as molecular determinants of BSIs.

Lines 679-683

To screen the virulence gene content present in the CC95 PAI and in the PAIs associated to each of the hybrid assemblies generated from the other K1-*cps* lineages (Table S4), we used amrfinder (version 3.10.18) with the flags `-plus` and `-organism` (indicating *Escherichia* as the taxonomic group)¹²³.

4. While benefits of K1 are proposed there are no discussion of the associated costs.

This is an excellent point raised by the reviewer. We now included a Discussion of the costs of K1 capsule production to *E. coli*. Please also see the response to reviewer 1 point 5.

Lines 478-488

Our analysis demonstrates that the K1-*cps* can be efficiently acquired by *E. coli*, and that K1 capsule expression provides protection against the human immune system. If K1-*cps* acquisition is so beneficial to *E. coli*, then the fact that 75% of all BSI *E. coli* isolates do not possess K1-*cps* genes suggests that there are costs to a bacterium associated with K1 capsule expression. The immediate costs of K1 capsule production are reduced growth rates, as the production of K1 polySia imposes a substantial metabolic burden⁸⁹, and an increased susceptibility to killing by bacteriophage that targets the K1 capsule⁹⁰. Additionally, capsule synthesis may cause a drastic reductions in plasmid conjugation⁸⁶, which could limit the acquisition of new traits required to maintain survival or competitiveness in the local environment, such as antimicrobial resistance genes which are usually disseminated through plasmid conjugation.

5. One of the underlying question that drives the present work, is whether the capsule would be properly expressed across background; There could be more arguments about why we do not expect it.

We thank the reviewer for this interesting question. Our analysis in Fig 5d shows that 98.6% of K1-*cps* positive *E. coli* isolates express detectable levels of the PolySia K1 capsule. This analysis includes isolates from 4 different phylogroups (A, B2, D and F) (Fig 5d), indicating that the K1 capsule is properly expressed across different genetic backgrounds. Our analysis also demonstrates heterogeneity in the K1 expression levels, indicating that additional factors regulate the amount of capsule expression in different genetic backgrounds. K1 PolySia is a polymer of α 2,8-linked N-acetylneuraminic acid (NeuNAc; Sialic acid), in which synthesis is dependent on the intracellular availability of NeuNAc (Whitfield 2006 DOI:10.1146/annurev.biochem.75.103004.142545). Thus, mechanisms that regulate NeuNAc levels also regulate K1 PolySia synthesis. Notably, NeuNAc can be acquired from the local environment via NanT, in addition to being produced endogenously from the K1-*cps* locus (Whitfield 2006 DOI:10.1146/annurev.biochem.75.103004.142545). In addition, due to the toxic effects of accumulation, NeuNAc is degraded by an aldolase called NanA. Thus, expression levels of K1-*cps* genes or *nanA* and/or *nanT* could contribute to the inter-strain variation in PolySia K1 capsule expression. We have described this in the results.

Lines 355-358

The heterogeneity in the K1 expression levels could have arisen from variation in the expression of genes for endogenous NeuNAc synthesis in the K1-*cps* locus, a NeuNAc uptake transporter called NanT (encoded by *nanT*) and/or a NeuNAc degradation enzyme called NanA (encoded by *nanA*)²⁷.

The reviewer raises an interesting point about when we expect the K1 capsule not to be expressed. Group 2 capsules, including K1, are expressed in a temperature-dependent manner (Cieslewicz & Vimr 1996 DOI:10.1128/jb.178.11.3212-3220.1996). Based on this, we expect that the K1 capsule is expressed in mammalian hosts due to high transcription at 37°C, but expression would be limited outside the body due to transcriptional repression at lower temperatures. In addition, data has emerged that shows that the expression of K1 capsule varies between body compartments. K1 expression is evidenced during growth in the gastrointestinal tract and blood compartments (Zelmer *et al.* 2008 DOI: 10.1099/mic.0.2008/017988-0), but is switched off during culture in urine (King *et al.* 2015 DOI:10.1128/IAI.00188-15) and during migration into the brain (Zelmer *et al.* 2008 DOI: 10.1099/mic.0.2008/017988-0). Though these findings suggest that there are benefits and costs to K1 capsule expression inside and outside mammalian hosts, further investigation

is required to understand the dynamics and functions of the K1 capsule expression during the pathway to infection. We have included this in the discussion.

Lines 488-492

The fact that K1 capsule expression is thermoregulated ^{27,91}, downregulated in the urine/bladder environment ⁹² and switched off in the brain environment ⁴⁰, is further suggestive that there are costs associated with K1 expression inside and outside of the mammalian host. These findings highlight the need for future studies that establish the dynamics and functions of K1 capsule expression on the pathway to causing infection.

I really appreciated the discussion that is humble and well-balanced. Overall it is a very solid study.

We appreciate and thank the positive words of the reviewer regarding the robustness of our study.

Reviewer #2

The manuscript titled “Evolutionary and Functional History of the Escherichia coli K1 Capsule” by Arredondo-Alonso et al. describes the role and evolution of the K1 capsule in infection by E. coli from extra intestinal sources. The authors rely on the reanalysis of ExPEC genomes from two large studies that have been deposited in the public databases. The work is focused on the presence and absence of the K1 capsule loci in the ExPEC isolates, but does not examine the prevalence in any of the diarrheagenic E. coli, which are also a significant source of blood stream infections, and may actually be the more common source, but are totally ignored in this study. The phylogenomic studies lack significant details which would allow a reader to be able to recreate the analyses presented. Evolutionary studies attempt to identify when the kps locus was acquired in these specific genomic backgrounds. These studies lack rigour and should be completed using statistics to support the conclusions. The authors spend a significant amount of time on the pathogenicity island associated with the kps loci, yet spend very little time describing what is within this actual PAI. Functional analyses indicated that the capsule was functional and could bind the ligand and prevent killing by complement. Overall, there are significant biases in the selection of the isolates in this study, there is a lack of statistical tests included in the analyses and the data presentation especially in the

supplemental data is lacking resulting in a lack of enthusiasm for this study and lack of confidence in the conclusions.

We thank the reviewer for their evaluation of our manuscript. As suggested, we have i) analyzed an association between diarrhoeagenic isolates and the K1 capsule loci by analyzing more than 5,000 diarrhoeagenic isolates with a pathotype defined and found no association, ii) emphasized that the NORM and BSAC studies were chosen as the basis of our analysis because their criteria for selection and inclusion of isolates, which provided us with a non-biased snapshot of the ExPEC population, iii) we have now substantially extended and provided all the statistics behind the dating models to show the robustness and confidence of the dating phylogenetic analyses. Below, we provide an extensive description of the new analyses and information incorporated in the manuscript following the evaluation of the reviewer. We have expanded our description of the PAI, as outlined in our response to Reviewer 1 point 3.

Major Comments

1. The K1 was only considered in the “most predominant extraintestinal lineages” which will result in a bias to previous and possibly more ancestral introductions. For example, the authors suggest that the CC95 isolates are the most successful clones, but the selection of the isolates is biased by the inclusion and examination of the genomes. These biases result in the conclusions of this study being significantly flawed and these flaws are not addressed in a meaningful way in the current study. The authors may suggest that they have not generated the data thus are not responsible for the biases, but they do have the ability to select any of the >50000 E. coli genomes that are currently in the public domain to address these biases.

We acknowledge that the selection criteria could have been better emphasized in the manuscript text. The NORM and BSAC collections were specifically selected for use in this study because these collections of BSI isolates were collated under specific and strict criteria to avoid selection bias. Both studies were longitudinal studies for which isolates causing BSI were collected and sequenced regardless of their antimicrobial resistance profile, clonal background or any genotype/phenotype of interest. This means they offer a representative survey of ExPEC clones that have circulated in comparable host

populations during the timespan of the studies. In contrast, the inclusion or selection from >50,000 *E. coli* genomes available in the public domain are biased towards lineages with a multi-drug resistant profile (mostly CC131 lineage) which would have severely misled the estimation of the prevalence of the K1-*cps* locus in the ExPEC population, as evident from the current study. For these reasons, data in the NORM and BSAC collections are suitable, whilst data from the public domain is unsuitable, to estimate the K1-*cps* prevalence in *E. coli* BSI populations. In line with this comment from the reviewer, we have modified the text detailing our selection criteria to improve clarity for a broad readership.

Lines 132-143

To estimate the prevalence of the K1-*cps* locus among ExPEC isolates, we assessed two unbiased longitudinal studies, NORM⁵⁰ and BSAC⁵¹, that characterized BSIs in Norway ($n = 3254$) and United Kingdom ($n = 1509$), respectively. As the BSI isolates in both studies were collected regardless of their clonal background, antimicrobial resistance profile or other bacterial phenotypic or genotypic characteristics, the NORM and BSAC collections offer a representative survey of BSI clones that have circulated in comparable host populations during the timespan of the studies. These datasets therefore provide a valuable platform to estimate the K1-*cps* prevalence in *E. coli* BSI populations. This population prevalence of the K1-*cps* locus among BSI isolates was estimated to be 24.0% and 22.9% for the NORM and BSAC collections, respectively (Table 1).

The reviewer rightfully pointed out that we did not estimate the K1-*cps* prevalence in diarrheagenic *E. coli* isolates. As previous literature has established that pathogenic intestinal isolates including those causing diarrhoea typically express capsules belonging to groups 1 or group 4 (Whitfield 2006 DOI:10.1146/annurev.biochem.75.103004.142545), we expected that the K1 capsule machinery would not be present in diarrheagenic isolates. As we agree that it is important for non-specialised readers to understand the association of the K1-*cps* to ExPEC, we have now assessed the distribution of K1-*cps* in other *E. coli* pathotypes using the genomic data that is now available for thousands of *E. coli* isolates. To do this, we screened the Horesh collection of *E. coli* genomes (DOI:10.1099/mgen.0.000499) which consists of over 10,000 genomes with a pathotype inferred. From this collection, we analysed all 5,236 diarrhoeagenic isolates comprising (i) enteropathogenic *E. coli* (EPEC), (ii) enterotoxigenic *E. coli* (ETEC), (iii) enterohaemorrhagic *E. coli* (EHEC), (iv) enteroaggregative *E. coli* (EAEC), (v) enteroinvasive *E. coli* (EIEC), (vi) diffusely adherent *E. coli* (DAEC) and (vii) adherent invasive *E. coli* (AIEC) for K1-*cps* positivity. We observed that only 5 of 5,236 isolates corresponding to 0.1% had the K1-*cps* locus. Therefore, we

conclude that the K1 capsule does not have a role in diarrhoeagenic diseases caused by *E. coli*. The analysis discarding the role of the K1-*cps* locus in diarrhoeagenic diseases is now presented in the results.

Lines 145-155

Group 2 capsules, including K1, are classically assumed to be expressed in *E. coli* isolates causing extraintestinal infections, but not in *E. coli* causing diarrhoeal diseases²⁷. To clarify whether the K1-*cps* is associated with any other *E. coli* pathotypes, we analyzed the Horesh et al. collection that consisting of a comprehensive, high-quality and pathotype-defined collection of *E. coli* genomes⁵². We specifically screened for the presence of the K1-*cps* locus in the 5,236 diarrhoeagenic isolates from the Horesh et al. collection that includes the pathotypes (i) enteropathogenic *E. coli* (EPEC), (ii) enterotoxigenic *E. coli* (ETEC), (iii) enterohaemorrhagic *E. coli* (EHEC), (iv) enteroaggregative *E. coli* (EAEC), (v) enteroinvasive *E. coli* (EIEC), (vi) diffusely adherent *E. coli* (DAEC) and (vii) adherent invasive *E. coli* (AIEC). We observed that only 0.1% (5/5236) of the diarrhoeagenic isolates carried the K1-*cps* locus, therefore discarding a role of the K1 capsule in diarrhoeagenic diseases.

2. Only ¼ of the BSI contains the K1 capsule – the authors have not demonstrated that this is clinically or statically significant. This is a significant gap in the importance of the K1 capsule in this group of pathogens. This is significant considering that the vast majority of these isolates are from blood. Additionally the date of 250 years ago indicated in line 190 is not supported by the Figure 2, as this figure is for the presence or absence of the PAI associated with the kps cluster, not the actual cluster itself. Other dates of 500 years ago are mentioned in the discussion, but not in the results. These dates are not well supported in the results, and the lack of statical methods included with these analyses makes these dates not well supported.

The clinical importance of K1 is well demonstrated in BSI and neonatal meningitis, as stated and referenced in the manuscript (Lines 78-80, 91-92). In addition, several groups have reported that K1 expression is associated with enhanced virulence in experimental models of infection, as stated in the manuscript. We have modified the wording on this second point, as we acknowledge the phrasing could be improved for broad readership.

Lines 97-100

In agreement with epidemiological links of K1 capsule to invasive human infections, experimental animal models using isogenic strains have revealed that K1 expression promotes stable gastrointestinal (GI) tract colonisation and promotes the development of invasive systemic infections by *E. coli*³⁹⁻⁴².

The fact that we find ~25% of a representative survey of BSIs to have the K1-*cps* locus is an epidemiological observation. Whether it is statistically significant or not could only be addressed against a specific null hypothesis. However, there is no prior information to suggest a particular population prevalence of K1 that should be tested in light of our data. Further, using a prevalence of zero would not be meaningful since we already know that K1-*cps* exists in the ExPEC population.

We thank the reviewer for their critique of Figure 2. We agree that the presentation of this Figure could be better aligned with the message of the results and the overall manuscript. As such, we have modified the Figure to map genomes with K1-*cps* present (red) and K1-*cps* absent (grey) as in Figure 3, and also plotted the presence or absence of the pathogenicity island (PAI) as a metadata block. Figure 2 now clearly displays that the acquisition of the K1-*cps* in CC95 is dated to 1768. The confidence and robustness of the dating model are explained and shown in the point below, and we have included the confidence intervals for all the dates given in Figure 2 in the figure legend.

3. The lack of any statistical tests associated with the dating in Figures 2 and 3 makes any conclusions made from this data suspect. Additionally, Figure 3 as presented is weak and unsupported and not well described in the results.

We thank the reviewer for pointing out that not all the statistics behind the dating models were provided in the manuscript. The dating models were computed using BactDating (Didelot *et al.* 2018 DOI:10.1093/nar/gky783) which uses a Bayesian framework for the inference of ancestor dating. The performance and robustness of the models are now given as a new Table (Table S2) providing the convergence of the Markov Chain Monte Carlo (MCMC) (1e8 iterations) as shown by i) the effective size of the parameters, (ii) the Gelman and Rubin's convergence diagnostic. In addition, each model was compared against a model with equal dates using the deviance information criteria (DIC) to show the significance of the temporal signal reported. We have now added clarification to the approach in the results.

Lines 207-217

To determine the origin of CC95 (tMRCA), we used BactDating that makes use of a Markov Chain Monte Carlo (MCMC) model to perform Bayesian inference and produce a dating phylogeny. We estimated the origin of CC95 (tMRCA) and thus estimated the introduction of the K1-*cps* locus to be

approximately around the year 1768 [95% CI, 1721-1806] (Fig. 2a). In Table S2, we confirmed the MCMC convergence of the model as shown by (i) the effective sample sizes of the parameters and (ii) the Gelman and Rubin's convergence diagnostic. In addition, we show a significant temporal signal by comparing the resulting model against a model with equal sampled dates using the deviance information criteria (DIC) (Table S2). Thus, the high-frequency of K1-*cps* in CC95 is readily explained by a single acquisition event that occurred approximately over 250 years ago that subsequently spread worldwide as a single clone.

Furthermore, we have included the 95% confidence intervals for all the dates given in Figures 2 and 3 (in the legends) as well as in the results at appropriate places in the text.

4. The isolate IDs provided in Supplementary Table 1 (assumed to be) does not match the isolate numbers provided in the ENA database for the provided accession numbers. The authors should ensure that these designations are consistent.

In Table S1, the isolate IDs match with the isolate names (column 'ID') provided in the BSAC (Kallonen *et al.* 2017 DOI:10.1101/gr.216606.116) and NORM (Gladstone *et al.* 2021 DOI:10.1016/S2666-5247(21)00031-8) studies. To facilitate their retrieval from the public databases, the column 'Accession' provides the reference of the short-reads publicly available in the ENA database. We have now clarified this in the new document describing each of the Supplementary Files included in the manuscript.

5. The authors mention increases in fitness in the abstract, introduction and discussion but have no actual measures of fitness. Mentions of fitness should be removed unless the authors want to complete these studies correctly. The same can be said for increasing antimicrobial resistance and invasiveness – none of these phenotypes are actually measured.

We thank the reviewer for pointing out that 'fitness' was incorrectly used, outside its usual interpretation in bacterial ecology and evolution. We have now accordingly reworded this on all previous occurrences.

6. A general lack of labeling of any of the supplemental documents makes the supplemental information difficult to navigate and essentially useless. An experienced group such as is assembled here has not actually seen and reviewed

this manuscript any detail, is asleep at the wheel or does not care enough to even look at these documents for clarity. Any of these options leads this reviewer to suggest that the remainder of the manuscript is suspect in terms of attention to detail. This may have something to do with the 25 authors in the author list, but only 11 authors have a listed contribution other than manuscript revision/ If the role of 14 authors was ONLY to review and revise the manuscript, these issues should not be present. The corresponding authors should review the rules for authorship to ensure that each of these named individuals actually meet the established criteria to be an author.

Though each of the supplemental documents is referred to in the manuscript with a clear definition of contents, we acknowledge that the supplemental information could be presented more clearly. To reduce confusion for readers, we have now provided a file with a summary and description of the supplemental information. In addition, we have simplified Supplementary Table S1.

All authors meet the established criteria of authorship.

Minor comments

1. The authors state that 8 genes are conserved and 6 are not in this gene cluster, but in Figure 1 it is not clear which are which – this should be clarified somewhere in the manuscript either in the figure legend or the text.

We believe the reviewer has missed the information that was provided in the legend of Figure 1.

Regions 1 and 3 (in green and purple, respectively) are responsible for capsule export and are shared among K antigens belonging to group 2 capsules. Region 2 (in yellow) is serotype-specific and involved in capsule biosynthesis.

2. Line 52 – define “pandemic lineages” this term has different meanings in today’s terms compared to previous eras.

Rapidly globally disseminated lineages of *E. coli* have been termed 'pandemic lineages' in the existing literature. However, to avoid misinterpretation we have removed this wording from the manuscript.

3. The authors should define that UTI are invasive infections. Many believe that these infections are not invasive.

We thank the reviewer for highlighting this point. Because many UTIs may not be invasive (i.e. asymptomatic bacteriuria or bladder infection (cystitis)), we have removed naming UTI alongside BSIs, pyelonephritis and meningitis from lines 77-80.

Although *E. coli* can produce around 80 distinct capsular chemotypes that are organised into four major groups ¹⁹, only a subset of these chemically distinct capsular types are associated with the capacity to cause invasive extraintestinal diseases; such infections include bloodstream infections (BSI), pyelonephritis and meningitis ²⁰.

4. Define "enhances *E. coli* fitness"

As explained above in response to Main Issue 5, 'fitness' is no longer used in the revised text.

5. In Supplemental Table 1 what is the difference between UTI and Urine?

We have modified designation of all isolates to "Urine".

REVIEWERS' COMMENTS

Reviewer #1 (Remarks to the Author):

The authors have properly answered to my comments.

I just think Figure 3 could be made nicer:

first the x axes erase the lowest branches of the phylogeny.

second the red dots are too big

third , it is not always clear to what the time line correspond, collaring in red the branches associated with the presence of the capsule could make that easier, dates could also be written below each phylogeny rather than at the bottom.